# HiST: A Hierarchical Sparse Transformer for Cross-Modal Spatial Transcriptomics Modeling

**Weiyi Wu** [1]  **Xinwen Xu** [2]  **Xingjian Diao** [1]  **Siting Li** [1]  **Zhi Wei** [3]  **Alma Andersson** [4]  **Jiang Gui** [1]

## Abstract

Spatial transcriptomics (ST) links gene expression with tissue morphology but remains expensive and low-throughput, motivating surrogates that infer expression from routine histology. Whole-slide H&E-to-ST inference pairs a gigapixel image with gene measurements at a sparse, irregular set of locations, making multiscale modeling challenging without incurring dense-grid overhead or quadratic token mixing. We propose HiST, a hierarchical sparse transformer that treats measured locations as a lattice-indexed sparse field and builds a dyadic encoder–decoder directly on the active tissue footprint. HiST combines sparse window attention for local geometric correspondence with resolution-changing operators for rapid multiscale context integration. For a fixed window size, the dominant runtime and memory scale with the number of observed locations rather than the dense slide area. To mitigate slide-specific acquisition variation, HiST adds a bottlenecked global conditioning pathway via a *slide calibration token* that summarizes slide-level context and conditions local representations. On a multi-organ benchmark spanning diverse tissues and acquisition sources, HiST improves predictive performance over recent baselines while reducing runtime and peak memory. Code is available at:
https://github.com/wwyi1828/HiST.

## 1. Introduction

Spatial transcriptomics links gene expression to tissue morphology by measuring transcriptomes with spatial coordinates, enabling molecular characterization of tissue organization (Ståhl et al., 2016; Moses & Pachter, 2022; Marx, 2021). By localizing molecular programs within tissue, Spatial transcriptomics (ST) has revealed spatially structured gene expression, microenvironmental niches, and long-range gradients that are invisible to bulk or dissociated assays (Moses & Pachter, 2022; Birk et al., 2025; Kueckelhaus et al., 2024; Aung et al., 2025; Andersson et al., 2021). However, ST assays remain expensive and low-throughput, limiting dataset scale and motivating computational surrogates that infer spatial expression from standard imaging data (Marx, 2021; Smith et al., 2024; Xie et al., 2023).

**H&E as a scalable surrogate.** Routine hematoxylin-and-eosin (H&E) histopathology is ubiquitous in clinical workflows, inexpensive to generate, and increasingly digitized into whole-slide images (WSIs) at scale (Dunn et al., 2024; Wu et al., 2023; 2026b). Accurate H&E-to-ST inference could turn these routine slides into a proxy molecular readout, enabling retrospective studies on large clinical archives and reducing reliance on expensive ST assays when they are impractical. Because morphology reflects underlying molecular programs, H&E contains rich latent molecular signals, and many ST platforms provide a co-registered histology image for localizing measurement locations (Chen et al., 2024a; Ståhl et al., 2016). In this work, we study H&E-to-ST inference: given ST coordinates and a co-registered H&E WSI, we predict a gene expression vector at each measured position.

Whole-slide inference is challenging because it requires mapping gigapixel images to high-dimensional gene expression vectors spanning thousands of genes, while supervision forms a sparse and irregular footprint. Accurate prediction requires multiscale tissue context, yet dense-grid backbones waste computation on background regions or padding when the tissue footprint is irregular, and global token mixing over WSIs is prohibitively expensive. A practical solution should therefore operate directly on the active tissue footprint and have dominant runtime and memory that scale primarily with the number of measured locations ($N$), while remaining robust to slide-specific acquisition variation such as staining and scanner differences (Schmitt et al., 2021; Millard et al., 2025). Recent iterative generative paradigms, including flow matching (Huang et al., 2025) and diffu-

---

[1]Dartmouth College [2]Mass General Hospital [3]New Jersey Institute of Technology [4]Genentech Inc.. Correspondence to: Weiyi Wu <weiyi.wu.gr@dartmouth.edu>.

*Proceedings of the 43rd International Conference on Machine Learning*, Seoul, South Korea. PMLR 306, 2026. Copyright 2026 by the author(s).

*Figure 1.* Overview of HiST. We extract patch features at measured tissue locations and arrange them on a coordinate-aligned sparse lattice. A spatial merging module progressively aggregates neighboring tokens to build compact multiscale representations, and a spatial recovery module with skip connections restores fine-grained features for per-spot prediction. Computation is applied only on observed tissue locations, keeping the dominant cost proportional to the number of spots.

sion (Zhu et al., 2025), can also be computationally intensive at whole-slide scale and large gene dimension due to multi-step inference.

**Limitations of fixed-range spatial coupling.** Prior H&E-to-ST models either (i) predict each location independently from its local patch (He et al., 2020; Chen et al., 2024b; Xie et al., 2023; Wang et al., 2025) or (ii) exchange information through fixed-range local interactions, such as graph message passing (Zeng et al., 2022) or neighborhood attention (Huang et al., 2025). While efficient and scalable, fixed-range coupling can make long-range tissue structure difficult to capture without stacking many layers, increasing both cost and sensitivity to neighborhood design.

**HiST: hierarchical sparse multiscale modeling.** To address these challenges, we propose HiST, a hierarchical sparse transformer for the ST data modeling. HiST is designed for whole-slide hierarchical modeling directly on the sparse tissue footprint defined by measured ST locations, which motivates our lattice-indexed sparse design. Figure 1 provides an overview. We focus on the spatial representation and backbone by casting the targets as a lattice-indexed sparse field and training with a standard supervised regression objective to keep the evaluation focused on this representation. This focus is complementary to alternative training paradigms, such as contrastive learning and iterative generative refinement: we show that an efficient backbone can already achieve strong accuracy under standard supervision, and these objectives could be layered on top of HiST in future work. HiST maps measured locations onto a sparse 2D lattice extracted from tissue patches, avoiding computation on background regions. It then builds a dyadic encoder–decoder hierarchy that alternates spatial coarsening and refinement, with skip connections that fuse features across resolutions. This design is analogous in spirit to a U-Net (Ronneberger et al., 2015), but operates directly on the sparse spatial domain.

To mitigate slide-specific acquisition shifts, we introduce a bottlenecked conditioning mechanism. Specifically, a distribution-aware slide calibration token attends to all locations to form a slide-level summary of the feature distribution, which is then broadcast back to condition local predictions. This provides a lightweight pathway for handling slide-specific acquisition variation under cross-slide training and evaluation.

**Contributions.** Our main contributions are:

- We introduce HiST, which maps measured locations onto a sparse 2D lattice and uses a dyadic encoder–decoder hierarchy to capture multiscale tissue context with dominant computation proportional to the number of observed locations.

- We propose a distribution-aware slide calibration token as a bottlenecked global conditioning pathway to mitigate slide-specific acquisition variation.

- We evaluate HiST on a multi-organ benchmark, matching or improving predictive performance while substantially reducing runtime and memory compared to recent baselines.

As shown in Tables 2 and 1, HiST matches or improves predictive performance while being about an order of magnitude faster at inference and over an order of magnitude lower in peak reserved memory than recent iterative baselines on our benchmark.

## 2. Related Work

**Computational tasks in spatial transcriptomics.** Spatial transcriptomics has driven a broad range of computational tools across complementary tasks. Graph-based ST models such as SpaGCN (Hu et al., 2021) integrate gene expression, spatial location, and histology to identify spatial domains and spatially variable genes. Multimodal generative models such as SPATIA (Kong et al., 2025) target prediction and generation of spatial cell phenotypes. Cross-modal integration frameworks such as SpatialMETA (Tian et al.,

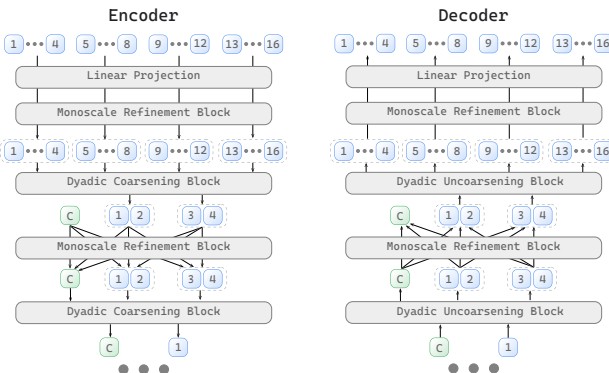

*Figure 2.* **Dyadic Encoder–Decoder Overview.** On the encoder side, shown on the left, local tokens are refined by a Monoscale Refinement Block conditioned on the slide calibration token $C_b$, followed by hierarchical grouping via a Dyadic Coarsening Block. The decoder, shown on the right, mirrors this process using Dyadic Uncoarsening and refinement blocks. Ellipses indicate repeated hierarchy levels, and token indices are shown for illustration only.

2025) combine spatial transcriptomics with metabolomics across samples. Downstream ST analysis tools such as NicheAgent (Dip & Zhang, 2025) use language-model-guided pipelines for zero-shot niche identification. These methods operate on already-measured ST; our work instead targets the upstream task of inferring ST from H&E at measured locations, reviewed next.

**H&E to ST prediction.** Computational pathology has expanded from slide-level prediction (Schmauch et al., 2020) to location-level models that infer spatially resolved molecular signals from histology. A common family of methods treats measured locations as i.i.d. samples (He et al., 2020; Chen et al., 2024b; Xie et al., 2023), predicting expression from local patch features via regression or retrieval in a joint image–expression embedding space (e.g., BLEEP). More recent context-aware approaches (Zeng et al., 2022; Xu et al., 2024; Jia et al., 2023) couple nearby locations, typically via message passing or attention over local neighborhoods in 2D Euclidean space. STFlow (Huang et al., 2025) casts ST prediction as a flow-matching generative process defined directly over the high-dimensional gene space, with iterative refinement and an E(2)-invariant local spatial-attention denoiser. Diffusion-based generators such as Stem (Zhu et al., 2025) provide another iterative paradigm by modeling expression via conditional denoising, but can be substantially more expensive due to multi-step sampling and scaling with the number of target genes. When information exchange is restricted to fixed local neighborhoods, long-range tissue context must be propagated through multiple rounds of local interaction and can be sensitive to neighborhood construction. Moreover, many designs do not explicitly exploit the approximately lattice-like sampling structure present in ST (Zhao et al., 2021; Walker et al., 2022).

**Hierarchical representation.** Hierarchical multiscale architectures are a core design principle in computer vision for capturing both fine-grained detail and long-range context. Canonical examples include U-Net-style encoder–decoder networks (Ronneberger et al., 2015; Lin et al., 2017) and hierarchical vision backbones (Liu et al., 2021; Woo et al., 2023; Hatamizadeh & Kautz, 2025; Wu et al., 2026a), which expand receptive fields through explicit resolution changes. Directly transferring dense-grid hierarchies to spatial transcriptomics is challenging because informative tissue occupies a sparse and irregular subset of the slide. As a result, many ST predictors rely on topological graph constraints rather than geometry-aligned lattice hierarchies: graphs handle irregular sampling but typically require deep stacks of local message passing to capture long-range structure. This motivates multiscale designs that operate directly on sparse tissue footprints while still leveraging the underlying 2D geometry.

**Slide acquisition variation.** Histology-based ST inference is affected by slide-specific nuisance variation, such as staining intensity, scanner differences, and cohort or batch effects, which can degrade cross-slide generalization (Schmitt et al., 2021; Howard et al., 2021). ST measurements can also exhibit pronounced batch effects, compounding these shifts under cross-slide training and evaluation (Millard et al., 2025). Common strategies include preprocessing and augmentation (Vahadane et al., 2016; Hoque et al., 2024). Complementary to these input-level strategies, another direction is to incorporate slide-level context as an explicit conditioning signal. We instantiate this idea with a slide calibration token: a learnable global token that reads out slide-level context by attending to all locations and broadcasts this summary back to condition local representations. This lightweight conditioning is conceptually reminiscent of learnable prompt/prefix tokens (Li & Liang, 2021; Jia et al., 2022) used to steer representations.

## 3. Method

### 3.1. Problem Formulation

**Data and Notation.** Spatial transcriptomics provides $N$ measurements $\{(\mathbf{s}_n, \mathbf{y}_n)\}_{n=1}^N$, where $\mathbf{s}_n \in \mathbb{R}^2$ is the 2D Euclidean coordinate and $\mathbf{y}_n \in \mathbb{R}^G$ is the non-negative expression vector over $G$ genes after fixed preprocessing. We treat these as sparse samples of an underlying 2D gene-expression field. Let $\mathcal{I} \in \mathbb{R}^{H \times W \times 3}$ denote the co-registered H&E whole-slide image. Each location is associated with an image patch extracted from $\mathcal{I}$, and a visual encoder yields a feature vector $\mathbf{x}_n \in \mathbb{R}^C$.

**Lattice-indexed Sparse Field Representation.** The observed locations typically occupy a sparse, non-rectangular subset of the slide due to tissue boundaries and missing

measurements. More generally, given coordinates $\mathbf{s}_n$, we define an indexing map $\pi(\mathbf{s}_n) = (i_n, j_n)$ onto a discrete sampling lattice $\mathcal{L} = \{0, \ldots, H' - 1\} \times \{0, \ldots, W' - 1\}$ at a chosen resolution. We represent gene expression as a sparse tensor field $\mathcal{Y} \in \mathbb{R}^{H' \times W' \times G}$, supported only on the set of observed indices $\mathcal{M} = \{(i_n, j_n)\}_{n=1}^N$. That is, $\mathcal{Y}[i_n, j_n] = \mathbf{y}_n$ for $(i_n, j_n) \in \mathcal{M}$, while locations outside $\mathcal{M}$ are *unlabeled* and not used for supervision, corresponding to out-of-tissue background regions or missing measurements. Analogously, we form a sparse feature field $\mathcal{X} \in \mathbb{R}^{H' \times W' \times C}$ with $\mathcal{X}[i_n, j_n] = \mathbf{x}_n$.

Our goal is to learn a function $F_\theta : (\mathcal{X}, \mathcal{M}) \to \widehat{\mathcal{Y}}$ that predicts expression at observed locations while leveraging spatial context on the lattice. We minimize a supervised loss restricted to the observed support:

$$\mathcal{L}_{\text{sup}} = \sum_{(i,j) \in \mathcal{M}} \ell\Big(\widehat{\mathcal{Y}}[i,j], \mathcal{Y}[i,j]\Big),$$

where $\ell(\cdot, \cdot)$ is a per-location regression loss such as squared error in log-expression space.

Representing ST as a sparse field on $\mathcal{L}$ provides several inductive biases. It enables weight sharing and directional kernels through relative offsets. It supports dyadic multiscale modeling. It also reduces background redundancy by focusing computation on observed tissue sites and their neighborhoods, rather than the entire slide area.

Overall, this view makes explicit that location-level ST targets behave as a sparse hyperspectral field on a 2D sampling lattice, rather than an unstructured point set. Unlike approaches that rely solely on fixed topological neighborhoods, our field formulation respects the underlying Euclidean geometry. Because tissue lies in continuous 2D space, this representation naturally supports geometric priors such as standard image augmentations and metric-aware relative offsets, which are less natural for purely topological neighborhood constructions.

### 3.2. Design Challenges and HiST Approach

Whole-slide H&E-to-ST inference must capture long-range context under tight compute budgets, scale to high-dimensional gene targets, and operate on sparse, irregular measurement footprints. Existing paradigms typically satisfy only part of this triad: independent per-location predictors ignore spatial context, fixed-range coupling propagates slowly, and dense multiscale backbones waste compute on background regions.

**HiST in a nutshell.** We propose a *Hierarchical Sparse Transformer* to resolve these tensions. First, we adopt lattice-indexed sparse field modeling that confines computation to the observed support $\mathcal{M}$, defined in Sec. 3.1. Second, we build a dyadic sparse hierarchy described in Secs. 3.5–3.6

that alternates sparse window attention with dyadic transitions, rapidly expanding receptive fields while keeping compute proportional to active tokens. Finally, we introduce a slide calibration token in Sec. 3.4 as a bottlenecked global conditioning pathway to robustify against acquisition shifts.

**Why hierarchy and sparsity?** Dyadic coarsening expands the spatial receptive field much faster than stacking local layers at a fixed resolution, and the token count shrinks by roughly $4\times$ per stage, making the total cost across stages a convergent geometric series. Combined with sparse window attention at each level described in Sec. 3.4, the overall compute across scales is dominated by the finest levels without padding the tissue bounding box. We provide a receptive-field analysis and a multiresolution view of these operators in Appendix B, including the Laplacian pyramid expansion in Eq. (6).

### 3.3. Overview

HiST alternates monoscale refinement blocks described in Sec. 3.4, which update tokens within each level via Dual-Path Attention, and dyadic transition operators described in Sec. 3.5, which coarsen/uncoarsen the active support to build a U-Net-like encoder–decoder.

**Multiscale lattice notation.** At level $\ell \in \{0, \ldots, L\}$ we work on a lattice $\mathcal{L}_\ell$ with active support $\mathcal{M}^{(\ell)} \subseteq \mathcal{L}_\ell$, where $\mathcal{M}^{(0)} = \mathcal{M}$. We denote the corresponding sparse feature field by $\mathcal{X}^{(\ell)}$ and let $N_\ell = |\mathcal{M}^{(\ell)}|$.

### 3.4. Monoscale Refinement Operators

We first define the core operator for updating features *within* a fixed lattice scale $\ell$. To simultaneously capture local morphology and correct for slide-wide acquisition shifts, we introduce the Dual-Path Attention mechanism. Let $X \in \mathbb{R}^{N_\ell \times d}$ denote the flattened tissue tokens at the current scale, let $C_b \in \mathbb{R}^{1 \times d}$ denote the slide calibration token defined in Sec. 3.6, and let $X' \in \mathbb{R}^{N_\ell \times d}$ denote the updated tissue tokens. We update the tissue representation via two parallel attention paths in a residual form:

$$X' = X + \text{Proj}\left(\underbrace{\text{W-MSA}(X)}_{\text{Local Spatial Context}} + \underbrace{\text{MCA}(X, C_b)}_{\text{Global Calibration}}\right). \quad (1)$$

Here W-MSA denotes sparse windowed self-attention on the active set, computed over both standard and shifted window neighborhoods, MCA denotes cross-attention between tissue tokens and the calibration token $C_b$, and $\text{Proj}$ denotes a learned output projection applied after summing the two attention outputs. Although Eq. (1) is written as a unit-weight sum, the relative contribution of the local and global paths is learned through their attention weights and projection

parameters. Eq. (1) decomposes the feature update into a local geometric refinement and a global distributional correction, ensuring that spatial mixing stays focused on tissue structure while slide-level biases are handled separately.

### 3.4.1. LOCAL SPATIAL CORRESPONDENCE

To mix information locally while respecting Euclidean offsets, we apply windowed multi-head self-attention on the sparse lattice. At each scale, we maintain an index map $\varphi^{(\ell)} : \mathcal{L}_\ell \to \{1, \ldots, N_\ell\} \cup \{\varnothing\}$ that maps each lattice site to a token index or to $\varnothing$ if empty. In practice, we build $\varphi^{(\ell)}$ as a lookup table over the tight lattice bounding box, enabling constant-time coordinate-to-token lookup during neighborhood construction. Using $\varphi^{(\ell)}$, we partition the lattice into non-overlapping windows of size $(2w) \times (2w)$ and gather the active tokens within each window $W$ while skipping unobserved sites. To enable cross-window information flow, we additionally apply a shifted partition with offset $w$ along both axes, so tokens near a window boundary can interact across it. This yields sparse local interaction neighborhoods: attention is computed only among token pairs that fall into the same standard or shifted window. Appendix A provides implementation details of window partitioning and how the index map is used to form these sparse neighborhoods. Let $\mathcal{W}^{(\ell)}$ and $\widetilde{\mathcal{W}}^{(\ell)}$ denote the standard and shifted window partitions, and let $Z_W$ be the active tokens gathered in window $W$. We apply Eq. (3) with interactions restricted to the sparse neighborhood induced by these partitions:

$$\text{W-MSA}(X) = \text{Attn}_{\mathcal{E}^{(\ell)}}(Q, K, V; B),$$

$$\mathcal{E}^{(\ell)} = \underbrace{\bigcup_{W \in \mathcal{W}^{(\ell)}} Z_W \times Z_W}_{\text{standard windows}} \cup \underbrace{\bigcup_{W \in \widetilde{\mathcal{W}}^{(\ell)}} Z_W \times Z_W}_{\text{shifted windows}}. \quad (2)$$

Here $\text{Attn}_{\mathcal{E}^{(\ell)}}$ denotes masked attention computed only over interaction pairs in $\mathcal{E}^{(\ell)}$. Accordingly, the dominant attention cost scales with window occupancy rather than the full token count: it is $\mathcal{O}\big(\sum_W |Z_W|^2\big) \leq \mathcal{O}\big(N_\ell (2w)^2\big) = \mathcal{O}(N_\ell)$ for fixed $w$, up to a constant factor for shifted windows, rather than $\mathcal{O}(N_\ell^2)$ as in dense global attention. Attention is computed with a learnable relative positional bias (RPB) $B$:

$$\text{Attn}(Q, K, V; B) = \text{Softmax}\left(\frac{QK^\top}{\sqrt{d}} + B\right) V, \quad (3)$$

where $Q, K, V$ are learned linear projections of the window tokens, and $B_{ab} = b(\Delta i_{ab}, \Delta j_{ab})$ encodes the relative lattice offset. This allows the model to learn direction-aware local interactions such as cell boundary continuity without relying on ad hoc positional encodings. Crucially, the bias depends on *relative* offsets, preserving translation equivariance on the lattice while allowing anisotropic local kernels—

a better match to tissue structures with consistent directional organization.

### 3.4.2. DISTRIBUTIONAL CALIBRATION

While the hierarchical backbone captures multiscale geometric structure, cross-slide generalization also requires accounting for slide-specific acquisition signatures. We therefore introduce an auxiliary global pathway that provides slide-level conditioning. The slide calibration token $C_b$ functions as a global low-rank information bottleneck. We instantiate $C_b$ as a learnable token shared across all slides, analogous to a CLS or prefix token, and within each forward pass it is refreshed by the attention update below to absorb the current slide's context. Specifically, tissue tokens attend to the current $C_b$ to receive a global conditioning signal:

$$\text{MCA}(X, C_b) = \text{Attn}(XW_Q, C_b W_K, C_b W_V; 0).$$

Simultaneously, we update $C_b$ by allowing it to attend to all tissue tokens and itself, thereby aggregating a slide-level summary:

$$C_b \leftarrow C_b + \text{Attn}(C_b W_Q^{(c)}, [X; C_b] W_K^{(c)}, [X; C_b] W_V^{(c)}; 0).$$

Here, $W$ terms denote learned linear projections. This factorization separates two roles: local spatial interaction is handled by lattice window attention, while $C_b$ serves as a low-bandwidth global channel that evolves to capture and condition on slide-wide distributional signatures.

### 3.5. Dyadic Transition Operators

We define operators that transfer information between lattice scales $\ell$ and $\ell + 1$, corresponding to the Laplacian-pyramid analysis operator $\mathcal{A}$ and synthesis operator $\mathcal{S}$. Both scale linearly in active-set size: each fine site maps to exactly one parent and each parent aggregates at most four children. Each dyadic coarsening step halves the lattice resolution along each axis, so receptive fields expand rapidly. Slide-wide context can be reached in approximately $O(\log N)$ hierarchy levels. Moreover, the number of active tokens typically shrinks by about $4\times$ per level ($N_{\ell+1} \approx N_\ell/4$), so for token-proportional operators the compute at coarser scales is roughly one quarter of that at the previous level.

### 3.5.1. COARSENING ANALYSIS

To integrate context and compress the lattice, we define a coarsening index mapping that assigns each active site $(i, j) \in \mathcal{M}^{(\ell)}$ to its dyadic parent $(\lfloor i/2 \rfloor, \lfloor j/2 \rfloor)$, yielding the coarsened support $\mathcal{M}^{(\ell+1)} = \{(\lfloor i/2 \rfloor, \lfloor j/2 \rfloor) : (i, j) \in \mathcal{M}^{(\ell)}\}$. For each parent site $(u, v)$, we gather its active children $\mathcal{C}_{u,v}$ and aggregate the dyadic $2 \times 2$ neighborhood on the lattice. Let $\delta \in \{0, 1\}^2$ index the four dyadic offsets and let $m_{u,v,\delta} \in \{0, 1\}$ indicate

whether the corresponding child site is active. Then

$$\mathbf{x}_{u,v}^{(\ell+1)} = \sum_{\delta \in \{0,1\}^2} m_{u,v,\delta} \, W_{\delta}^{(\ell)} \, \mathbf{x}_{2u+\delta_1,\, 2v+\delta_2}^{(\ell)}, \quad (4)$$

where $W_{\delta}^{(\ell)}$ are learned weights shared across lattice sites.

### 3.5.2. UNCOARSENING SYNTHESIS

The inverse operator recovers features at level $\ell$ from level $\ell + 1$ using an uncoarsening index mapping that records, for each parent site $(u, v) \in \mathcal{M}^{(\ell+1)}$, the corresponding set of active child sites $\mathcal{C}_{u,v} \subseteq \mathcal{M}^{(\ell)}$. We employ a sparse patch-expanding operator that maps each parent token $\mathbf{x}_{u,v}^{(\ell+1)}$ back to its children locations in $\mathcal{C}_{u,v}$:

$$\widetilde{\mathcal{X}}_{\mathrm{dec}}^{(\ell)} = \mathrm{Up}^{(\ell)}\Big( \mathcal{X}_{\mathrm{dec}}^{(\ell+1)}, \mathcal{M}^{(\ell+1)} \Big).$$

This synthesis step restores the spatial resolution required for fine-grained predictions. In practice, we instantiate $\uparrow$ by first applying an MLP that expands channels by 4, and then applying a channel-to-space rearrangement, i.e., pixel shuffle. Concretely, for each parent token,

$$\begin{aligned} \mathbf{z}_{u,v} &= \mathrm{MLP}^{(\ell)}\Big( \mathbf{x}_{u,v}^{(\ell+1)} \Big) \in \mathbb{R}^{4d}, \\ \{\mathbf{z}_{u,v,\delta}\}_{\delta \in \{0,1\}^2} &= \mathrm{reshape}(\mathbf{z}_{u,v}). \end{aligned} \quad (5)$$

and we scatter $\mathbf{z}_{u,v,\delta}$ to child coordinate $(2u + \delta_1, 2v + \delta_2)$, keeping only those children that lie in $\mathcal{M}^{(\ell)}$.

### 3.6. HiST Architecture

We now assemble these operators into the full hierarchical model.

#### 3.6.1. INITIALIZATION

We initialize the slide calibration token as a learnable vector shared across slides, $C_b = \mathbf{c}_0 \in \mathbb{R}^{1 \times d}$. During inference and training, $C_b$ is updated by the refinement blocks in Sec. 3.4 through attention over tissue tokens, producing a slide-specific summary that conditions local representations.

#### 3.6.2. HIERARCHICAL ENCODER

The encoder consists of $L$ stages. Each stage $\ell$ first applies a stack of Monoscale Refinement Operators in Sec. 3.4 to process features $\mathcal{X}^{(\ell)}$ conditioned on $C_b$. It then applies the Coarsening Analysis to produce the next level features $\mathcal{X}^{(\ell+1)}$. Features $\mathcal{X}_{\mathrm{enc}}^{(\ell)}$ are cached for lateral connections.

#### 3.6.3. HIERARCHICAL DECODER

The decoder reverses this process. Starting from the deepest level, each stage $\ell$ applies the Uncoarsening Synthesis operator $\uparrow$ to features from $\ell + 1$. To re-inject high-frequency

*Table 1.* **Computational Efficiency and Scalability.** We report throughput, measured as time per slide, and peak memory usage. Memory is reported at the **maximum feasible batch size** $N$ for each method. Parentheses report memory per 1k spots (GB/1k), normalized by the peak number of spots in training $N$.

| Method | Time (s/slide) | | Peak Memory Usage (GB) | |
|---|---|---|---|---|
| | Train | Infer | Allocated | Reserved |
| ST-Net | 6.51 | 2.02 | 4.72 (147.5) | 8.39 (262.2) |
| BLEEP | 9.54 | 9.90 | 10.65 (83.2) | 12.45 (97.3) |
| STFlow | 2.94 | 0.76 | 11.94 (1.8) | 46.37 (7.1) |
| HiST | **0.10** | **0.07** | **1.21 (0.3)** | **1.38 (0.4)** |

spatial details lost during coarsening, we employ a Lateral Context Fusion operation. This mirrors the role of the Laplacian pyramid expansion in Eq. (6) in App. B: the upsampled pathway carries coarse information while the lateral features provide same-scale detail. We fuse the upsampled features with the cached encoder features:

$$\mathcal{Z}^{(\ell)} = \left[ \underbrace{\widetilde{\mathcal{X}}_{\mathrm{dec}}^{(\ell)}}_{\text{Upsampled coarse context}} \;\big\|\; \underbrace{\mathcal{X}_{\mathrm{enc}}^{(\ell)}}_{\text{Lateral fine details}} \right],$$

where $\|$ denotes feature concatenation. The fused features are then processed by the refinement block:

$$\mathcal{X}_{\mathrm{dec}}^{(\ell)} = \Phi^{(\ell)}\Big( \mathcal{Z}^{(\ell)}; C_b \Big).$$

Finally, a linear head predicts log-expression $\widehat{\mathcal{T}}$.

### 3.7. Training objective

To enforce geometric robustness, we leverage our continuous field formulation by applying random planar transforms (e.g., rotation, shear) and random dropout to the coordinates $\mathbf{s}_n$ during training. The resulting transformed support $\widetilde{\mathcal{M}}$ serves as the domain for all sparse operators. This strategy mirrors standard image augmentation, keeping the computational footprint unchanged while preventing overfitting to fixed orientations.

We train in log-expression space. Let $\mathcal{T} = \log(\mathcal{Y} + 1)$, and let the model predict $\widehat{\mathcal{T}}$ at observed locations. We optimize the model using mean squared error, averaged over observed sites $N$ and genes $G$:

$$\mathcal{L}_{\mathrm{mse}} = \frac{1}{NG} \sum_{(i,j) \in \mathcal{M}} \left\| \widehat{\mathcal{T}}[i,j] - \mathcal{T}[i,j] \right\|^2.$$

## 4. Experiments

### 4.1. Experimental Setup

**Data and Protocols.** We evaluate whole-slide H&E-to-ST prediction on a large-scale multi-organ benchmark with

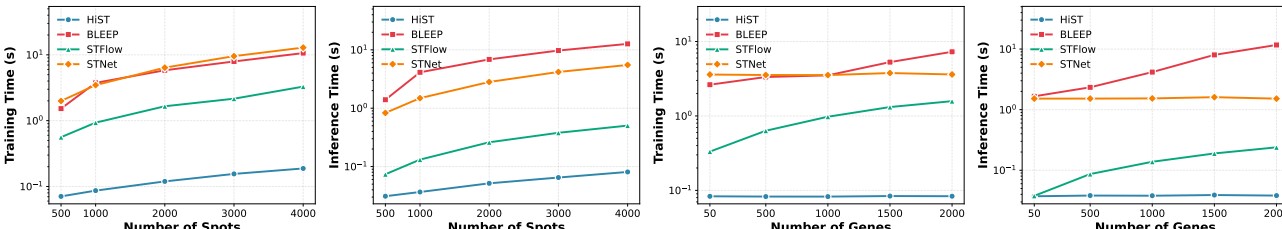

*Figure 3.* **Scalability across spatial and output dimensions.** From left to right, we plot train/spots, infer/spots, train/genes, and infer/genes; all axes are log-scale. All measurements are conducted on the same hardware.

HEST (Jaume et al., 2024) spanning 11 datasets (Table 2). The benchmark also includes data across multiple ST platforms, including 10x Visium, 10x Xenium, and legacy Spatial Transcriptomics. For each dataset, we employ a strict 4-fold cross-validation protocol where folds are split by slide ID. This ensures that all evaluation is performed on unseen tissue sections that were not seen during training, rigorously testing the model's ability to generalize rather than memorize. Furthermore, the benchmark spans datasets from diverse sources with varying protocols, introducing significant domain heterogeneity that makes the task considerably more challenging than single-source benchmarks. Preprocessing and dataset composition details (including slide IDs) are provided in App. A.1.

**Baselines and Metrics.** We compare HiST against STNet (He et al., 2020), BLEEP (Xie et al., 2023), UNI (Chen et al., 2024b), and STFlow (Huang et al., 2025). We focus on STFlow as a representative scalable generative baseline, as other diffusion-based methods like Stem (Zhu et al., 2025) are computationally prohibitive for whole-slide evaluation with $G = 2000$ genes. We optimize mean squared error (MSE) in log-expression space and report Coefficient of Determination ($R^2$) and Pearson Correlation Coefficient (PCC). We also report per-slide wall-clock time and peak GPU memory (Table 1). Training hyperparameters and hardware details are provided in App. A.

### 4.2. Comparison with Baselines

We compare HiST to baselines spanning three paradigms for H&E-to-ST inference: one-shot predictors, retrieval-based models, and iterative generative models. Table 2 summarizes performance across tissue types. Across datasets, HiST attains the best or tied-best $R^2$ and PCC among the compared methods and achieves the best average performance. On average, HiST improves $R^2$ from 0.27 to 0.34 (+0.07, +26%) and PCC from 0.54 to 0.59 (+0.05, +9%) compared to a recent strong spatially-aware baseline, STFlow. We note that absolute values are modest: morphology only partially reflects the transcriptome and ST measurements are noisy and sparse, which imposes a practical ceiling on predictive accuracy.

Qualitatively, position-independent predictors capture local texture but have limited mechanisms to integrate long-range tissue context, while STFlow models within-slide dependencies through iterative refinement with local interactions. HiST integrates multiscale context on the active tissue footprint via a sparse encoder–decoder hierarchy, keeping the dominant cost proportional to the number of observed locations.

**Where the accuracy gains come from.** HiST improves prediction by combining local geometric correspondence with rapid multiscale context integration: sparse window attention (with positional terms) captures local structure, while dyadic coarsening/uncoarsening with skip connections expands receptive fields and fuses information across scales. Ablations confirm the contributions of these components and the slide calibration token, with consistent drops when removing coordinate augmentation, positional terms, multiscale fusion, downsample–upsample transitions, or the calibration token (Table 3).

### 4.3. Computational Efficiency and Scalability

Table 1 reports average training and inference time per slide, along with peak memory usage. Figure 3 complements this snapshot by evaluating scaling with the number of spots $N$ and the output dimension $G$. Since the maximum feasible batch size $N$ differs across methods, we additionally report memory per 1k spots to contextualize peak memory numbers.

**Where the efficiency comes from.** HiST's efficiency follows from its sparse operator design: window attention computes interactions only among tokens that co-occur in the same (shifted) window, and dyadic coarsening reduces the number of active tokens by roughly $4\times$ per stage, so the total compute across scales is dominated by the finest resolutions.

**Predictive performance–efficiency trade-off.** Iterative refinement and retrieval can improve modeling flexibility but increase inference cost. As shown in Table 1, STFlow requires 0.76 s/slide inference time with 46.37 GB peak reserved memory (7.1 GB/1k), while HiST runs in 0.07 s/slide

*Table 2.* Quantitative comparison across organs and datasets. Dataset sources are shown in parentheses. We report $R^2$ (↑) and Pearson Correlation Coefficient (PCC, ↑) as mean ± std over 4 cross-validation folds on the top 2,000 highly variable genes (HVGs) when available; for Lung (NCBI Geo), we use all 541 available genes. Non-positive $R^2$ values are denoted as ≤ 0.

| Dataset | ST-Net | | BLEEP | | UNI | | STFlow | | HiST | |
|---|---|---|---|---|---|---|---|---|---|---|
| | $R^2$ (↑) | PCC (↑) | $R^2$ (↑) | PCC (↑) | $R^2$ (↑) | PCC (↑) | $R^2$ (↑) | PCC (↑) | $R^2$ (↑) | PCC (↑) |
| Brain (NCBI Geo) | $0.22_{\pm0.14}$ | $0.50_{\pm0.16}$ | $0.03_{\pm0.19}$ | $0.46_{\pm0.16}$ | $0.14_{\pm0.09}$ | $0.48_{\pm0.12}$ | $\mathbf{0.23}_{\pm0.11}$ | $0.50_{\pm0.08}$ | $\mathbf{0.23}_{\pm0.14}$ | $\mathbf{0.54}_{\pm0.19}$ |
| Colon (Other) | $0.04_{\pm0.03}$ | $0.31_{\pm0.05}$ | $0.10_{\pm0.10}$ | $0.34_{\pm0.17}$ | $0.02_{\pm0.01}$ | $0.19_{\pm0.09}$ | $0.11_{\pm0.06}$ | $0.42_{\pm0.06}$ | $\mathbf{0.31}_{\pm0.15}$ | $\mathbf{0.55}_{\pm0.15}$ |
| Heart (Other) | ≤ 0 | $0.09_{\pm0.10}$ | $0.20_{\pm0.06}$ | $0.49_{\pm0.04}$ | $0.06_{\pm0.04}$ | $0.27_{\pm0.08}$ | $0.18_{\pm0.08}$ | $0.50_{\pm0.04}$ | $\mathbf{0.22}_{\pm0.09}$ | $\mathbf{0.52}_{\pm0.08}$ |
| Kidney (NCBI Geo) | $0.12_{\pm0.02}$ | $0.39_{\pm0.05}$ | $0.13_{\pm0.06}$ | $0.40_{\pm0.05}$ | $0.14_{\pm0.04}$ | $0.40_{\pm0.06}$ | $0.11_{\pm0.04}$ | $\mathbf{0.43}_{\pm0.04}$ | $\mathbf{0.19}_{\pm0.08}$ | $0.43_{\pm0.10}$ |
| Liver (NCBI Geo) | $0.34_{\pm0.18}$ | $0.65_{\pm0.18}$ | $0.32_{\pm0.20}$ | $0.58_{\pm0.13}$ | $0.22_{\pm0.06}$ | $0.55_{\pm0.10}$ | $0.39_{\pm0.25}$ | $0.62_{\pm0.25}$ | $\mathbf{0.41}_{\pm0.22}$ | $\mathbf{0.66}_{\pm0.21}$ |
| Lung (Other) | $0.02_{\pm0.02}$ | $0.30_{\pm0.08}$ | $0.26_{\pm0.15}$ | $0.60_{\pm0.18}$ | $0.02_{\pm0.01}$ | $0.38_{\pm0.19}$ | $0.37_{\pm0.12}$ | $0.61_{\pm0.10}$ | $\mathbf{0.38}_{\pm0.17}$ | $\mathbf{0.65}_{\pm0.12}$ |
| Prostate (Mendeley Data) | ≤ 0 | $0.16_{\pm0.13}$ | $0.21_{\pm0.10}$ | $0.50_{\pm0.19}$ | $0.10_{\pm0.05}$ | $0.40_{\pm0.10}$ | $0.31_{\pm0.15}$ | $0.57_{\pm0.17}$ | $\mathbf{0.35}_{\pm0.16}$ | $\mathbf{0.61}_{\pm0.17}$ |
| Skin (NCBI Geo) | ≤ 0 | $0.15_{\pm0.17}$ | $0.21_{\pm0.02}$ | $0.49_{\pm0.03}$ | $0.09_{\pm0.03}$ | $0.33_{\pm0.06}$ | $0.21_{\pm0.03}$ | $0.51_{\pm0.02}$ | $\mathbf{0.31}_{\pm0.10}$ | $\mathbf{0.57}_{\pm0.07}$ |
| Uterus (NCBI Geo) | $0.11_{\pm0.09}$ | $0.40_{\pm0.08}$ | $0.01_{\pm0.10}$ | $0.32_{\pm0.13}$ | $0.09_{\pm0.08}$ | $0.32_{\pm0.12}$ | $0.11_{\pm0.08}$ | $0.34_{\pm0.12}$ | $\mathbf{0.13}_{\pm0.09}$ | $\mathbf{0.41}_{\pm0.11}$ |
| Breast (Spatial Research) | $0.17_{\pm0.04}$ | $0.45_{\pm0.08}$ | $0.55_{\pm0.09}$ | $0.75_{\pm0.05}$ | $0.25_{\pm0.12}$ | $0.52_{\pm0.13}$ | $0.44_{\pm0.08}$ | $0.68_{\pm0.05}$ | $\mathbf{0.69}_{\pm0.06}$ | $\mathbf{0.84}_{\pm0.04}$ |
| Lung (NCBI Geo) | $0.49_{\pm0.05}$ | $0.71_{\pm0.04}$ | $0.54_{\pm0.03}$ | $0.74_{\pm0.02}$ | $0.41_{\pm0.03}$ | $0.66_{\pm0.03}$ | $0.55_{\pm0.04}$ | $\mathbf{0.75}_{\pm0.02}$ | $\mathbf{0.56}_{\pm0.07}$ | $\mathbf{0.75}_{\pm0.05}$ |
| **Average** | $0.14_{\pm0.16}$ | $0.37_{\pm0.20}$ | $0.23_{\pm0.19}$ | $0.52_{\pm0.14}$ | $0.14_{\pm0.12}$ | $0.41_{\pm0.14}$ | $0.27_{\pm0.14}$ | $0.54_{\pm0.13}$ | $\mathbf{0.34}_{\pm0.17}$ | $\mathbf{0.59}_{\pm0.13}$ |

*Table 3.* **Component ablations on Brain (NCBI Geo).** Each block ablates one HiST component while holding the others at their default setting.

| Setting | $R^2$ ↑ | PCC ↑ |
|---|---|---|
| Full model | $0.228_{\pm0.13}$ | $0.542_{\pm0.18}$ |
| *Coordinate Augmentation* | | |
| None | $0.149_{\pm0.16}$ | $0.500_{\pm0.18}$ |
| Random Mirror Only | $0.153_{\pm0.17}$ | $0.503_{\pm0.18}$ |
| Random Shear Only | $0.156_{\pm0.17}$ | $0.504_{\pm0.18}$ |
| Random Rotate Only | $0.176_{\pm0.16}$ | $0.508_{\pm0.18}$ |
| Random Drop Only | $0.184_{\pm0.15}$ | $0.526_{\pm0.19}$ |
| *Positional Encoding: Spatial Correspondence* | | |
| None | $0.131_{\pm0.16}$ | $0.521_{\pm0.18}$ |
| RoPE | $0.153_{\pm0.14}$ | $0.522_{\pm0.17}$ |
| ALiBi | $0.132_{\pm0.16}$ | $0.522_{\pm0.18}$ |
| *Skip Connections: Multi-scale Fusion* | | |
| None | $0.197_{\pm0.10}$ | $0.527_{\pm0.19}$ |
| Additive | $0.213_{\pm0.14}$ | $0.536_{\pm0.20}$ |
| *Downsample-Upsample* | | |
| None | $0.213_{\pm0.15}$ | $0.517_{\pm0.18}$ |
| *Sparse Window Partition* | | |
| No shifted windows | $0.217_{\pm0.14}$ | $0.531_{\pm0.18}$ |
| *Slide Calibration Token* | | |
| None | $0.185_{\pm0.17}$ | $0.509_{\pm0.18}$ |

trade-off when scaling retrieval.

Iterative generative models require multiple refinement steps, so inference time scales with the number of steps. Moreover, STFlow performs refinement directly in the $G$-dimensional gene space: the intermediate states updated across refinement steps are $G$-dimensional gene vectors, so per-step compute and activation memory scale with $G$. In our benchmark with $G = 2000$, STFlow reaches $46.37\,\mathrm{GB}$ peak reserved memory.

### 4.4. Ablation Studies

We ablate key components corresponding to HiST's geometric and multiscale inductive biases. Specifically, we ablate coordinate augmentation and positional terms for spatial correspondence (Sec. 3.4), multiscale fusion (skip connections and downsample–upsample transitions), and the slide calibration token. Unless specified, we keep the default settings and change one factor at a time. We report mean ± std over four folds on Brain (NCBI Geo).

**Geometric inductive biases.** Coordinate-space augmentations perturb the measurement coordinates $\{\mathbf{s}_n\}$ while keeping the patch-derived features $\{\mathbf{x}_n\}$ fixed. The gains in Table 3 therefore suggest that HiST leverages spatial geometry and relative arrangement through its lattice-indexed operators, rather than relying solely on per-location appearance. Consistently, removing relative positional terms degrades predictive performance; we also evaluate standard attention positional schemes, RoPE and ALiBi (Su et al., 2024; Press et al., 2022), but they underperform our default positional terms (Table 3). We observe similar drops when removing skip connections or disabling downsample–upsample transitions, which supports the value of multiscale fusion (Table 3).

with $1.38\,\mathrm{GB}$ peak reserved memory ($0.4\,\mathrm{GB/1k}$). HiST maintains strong $R^2$ and PCC while substantially reducing both runtime and memory (Table 2).

BLEEP is retrieval-based and performs similarity search against a reference bank in an embedding space at inference time. This introduces an additional cost component beyond a single forward pass: inference time scales with the number of query spots and the size of the reference bank, and the memory footprint scales with the bank unless approximate indexing is used. This creates a practical accuracy–latency

# 5. Limitations and Future Directions

**Limitations.** HiST is designed for supervised H&E-to-ST inference at measured locations, and several limitations follow from this setting. First, training relies on paired histology–ST data with consistent preprocessing and normalization. The slide calibration token is intended to moderate, not eliminate, the effect of acquisition variation; distribution shift across cohorts, scanners, stains, or protocols may still degrade performance. Second, the sparse lattice representation depends on the choice of discretization resolution and on accurate coordinate registration between histology image and ST locations. Misalignment or systematic missingness could affect neighborhood structure and prediction quality. Third, current absolute accuracy is not yet sufficient for fine-grained clinical use; downstream clinical validation would require larger and more diverse paired ST cohorts than are publicly available today. Fourth, predictive accuracy is uneven across granularities. ST measurements are sparse and zero-inflated at the level of individual genes, so per-gene rank-level recovery remains constrained for many low-expression genes, while coarse-grained spatial organization (e.g., domain-level structure recovered by downstream clustering) is captured more reliably. Use cases that depend on accurate gene-by-gene profiles should be interpreted with this granularity gap in mind.

**Future Directions.** The hierarchical sparse backbone also opens several directions for future work. First, uncertainty quantification could be incorporated via test-time computing or ensembling strategies such as Monte Carlo dropout, which are particularly feasible given the efficiency of our inference pipeline. Second, by continuing the uncoarsening or upsampling process, HiST could potentially support slide-level super-resolution and predict expression maps at resolutions finer than the training data. Third, HiST's efficiency makes it a suitable backbone for scaling to pan-cancer foundation models or supporting computationally intensive iterative generative paradigms on whole slides.

# Acknowledgment

This study is supported by the Department of Defense grant HT9425-23-1-0267.

# Impact Statement

This paper develops models that predict spatial gene expression at measured locations from H&E histology and ST coordinates. If reliable, such models could reduce the cost of exploratory analyses, support hypothesis generation, and help scale studies where sequencing every slide is impractical. However, predicted expression is an inference rather than a measurement, and using these predictions for clinical decision-making without rigorous validation may introduce harm. Model performance may vary across tissue types, cohorts, and acquisition settings; spurious correlations and batch effects could also amplify biases if deployment populations differ from training data. We therefore view these methods as complementary tools for research workflows and emphasize the need for careful evaluation, transparency about uncertainty, and responsible use.

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

# A. Implementation Details

## A.1. HEST-1k benchmark details

We follow the preprocessing and evaluation pipeline provided by HEST (Jaume et al., 2024). Our benchmark includes data across multiple ST platforms, including 10x Visium, 10x Xenium, and legacy Spatial Transcriptomics.

**Preprocessing and evaluation.**   Gene filtering/normalization, patch extraction, and coordinate alignment follow the default HEST-1k preprocessing pipeline unless stated otherwise. We evaluate on the top 2,000 highly variable genes (HVGs) and report Coefficient of Determination ($R^2$) and Pearson Correlation Coefficient (PCC) as mean $\pm$ std over 4 cross-validation folds. All metrics are computed using the benchmark's reporting script under the default settings. Unless stated otherwise, we operate on precomputed patch embeddings and apply coordinate-only augmentations without re-encoding image patches.

**Cross-dataset harmonization.**   When training or evaluating across multiple sources, we ensure that molecular and morphology measurements are paired and represented in a consistent feature space. We (i) harmonize gene identifiers by stripping dataset-specific prefixes/version suffixes and mapping Ensembl IDs to gene symbols when needed, (ii) aggregate duplicated genes/barcodes by summation, (iii) restrict the gene space to the shared set across the slides involved and reindex each slide accordingly, and (iv) match spots between the expression matrix and patch data by barcode, keeping only the intersection for downstream processing.

*Table 4.* HEST-1k slide IDs for the datasets used in Table 2.

| Dataset | Slide IDs (#slides) |
|---|---|
| Brain (NCBI GEO) | NCBI628–NCBI641 (14) |
| Colon (Other) | MISC33–MISC73 (41) |
| Heart (Other) | MISC101–MISC136, MISC138–MISC142 (41) |
| Kidney (NCBI GEO) | NCBI538–NCBI540, NCBI562–NCBI568, NCBI599, NCBI692–NCBI714 (34) |
| Liver (NCBI GEO) | NCBI642–NCBI643, NCBI672–NCBI675, NCBI826–NCBI833 (14) |
| Lung (Other) | MISC13–MISC32 (20) |
| Prostate (Mendeley Data) | MEND59–MEND62, MEND139–MEND154, MEND156–MEND162 (27) |
| Skin (NCBI GEO) | NCBI460–NCBI498, NCBI503–NCBI510, NCBI515–NCBI526 (59) |
| Uterus (NCBI GEO) | NCBI573–NCBI576, NCBI811–NCBI821 (15) |
| Breast (Spatial Research) | SPA51–SPA154 (104) |
| Lung (NCBI GEO) | NCBI534–NCBI537 (4) |

**Dataset references.**   Dataset sources include Brain (Donson et al., 2022), Kidney (Lake et al., 2023; Canela et al., 2023; Villacampa et al., 2021; Ferreira et al., 2021), Liver (Matchett et al., 2024; Andrews et al., 2024; Giraud et al., 2022), Skin (Schäbitz et al., 2022), Uterus (Barrozo et al., 2023), and Lung (Madissoon et al., 2023) from NCBI GEO; Prostate (Erickson et al., 2022; Mirzazadeh et al., 2023) from Mendeley Data; Breast (He et al., 2020; Andersson et al., 2021) from Spatial Research; and Colon (Chen et al., 2021), Heart (Kanemaru et al., 2023), and Lung (Madissoon et al., 2023) from other repositories.

## A.2. Sparse lattice representation

We store active tissue sites $\mathcal{M} = \{(i_n, j_n)\}_{n=1}^N$ as an index list and maintain a feature table $X \in \mathbb{R}^{N \times C}$ aligned with this list. This decouples computation from the dense bounding box $H' \times W'$ and ensures that the dominant cost scales with the number of observed sites and their local neighborhood occupancy. When lattice operators require neighborhood grouping, we additionally build an *index map* $\varphi : \mathcal{L} \to \{1, \ldots, N\} \cup \{0\}$ over the tight lattice bounding box, storing the token id at each occupied site and 0 for empty sites. This map serves as a lightweight lookup table for forming sparse local interaction sets without introducing background tokens into attention.

## A.3. Window partitioning on the active set

For windowed attention at scale $\ell$, tokens interact within local windows. Conceptually, each active site $(i, j) \in \mathcal{M}^{(\ell)}$ is assigned to a window key by integer division; in practice, we realize window partitioning via scatter/gather operations on an integer index map. We use windows of side length $2w$ and stride $2w$, and we optionally apply a shifted partition with offset

---

**Algorithm 1** Build a sparse window rulebook from an index map (vectorized; high level)

---

**Require:** Active sites $\mathcal{M}^{(\ell)}$, window half-size $w$, optional shift offset $s$
**Ensure:** Rulebook $\mathcal{E}$ (and relative offsets) for window attention
1: $\varphi^{(\ell)} \leftarrow \text{ScatterIndexMap}(\mathcal{M}^{(\ell)})$ {lookup table over the tight bounding box; 0 for empty}
2: $W \leftarrow \text{ExtractWindows}(\varphi^{(\ell)}, w, s)$ {batched window extraction; keep non-empty}
3: $Z \leftarrow \text{Flatten}(W)$ {token-id vectors per window}
4: $\mathcal{E} \leftarrow \text{PairAndFilter}(Z)$ {broadcast pairs and mask out 0 entries and self-pairs}
5: $\mathcal{E} \leftarrow \text{Dedup}(\mathcal{E})$ and compute $(\Delta i, \Delta j)$ {optional}
6: $(\mathcal{E}, \Delta i, \Delta j)$

---

**Algorithm 2** Dyadic coarsening and MLP-based uncoarsening (vectorized; high level)

---

**Require:** Active sites $\mathcal{M}^{(\ell)}$ and aligned features $X^{(\ell)} \in \mathbb{R}^{N \times d}$
**Ensure:** Parent sites $\mathcal{M}^{(\ell+1)}$, coarsened features $X^{(\ell+1)}$, upsampled features $\widetilde{X}^{(\ell)}$
1: {Coarsen: gather up to four children per $2 \times 2$ parent cell and aggregate}
2: $P \leftarrow \lfloor \mathcal{M}^{(\ell)}/2 \rfloor$ {parent coordinates, vectorized}
3: $(\mathcal{M}^{(\ell+1)}, \pi) \leftarrow \text{UniqueWithInverse}(P)$ {unique parents and child→parent map}
4: $\sigma \leftarrow \text{DyadicSlot}(\mathcal{M}^{(\ell)}) \in \{1, 2, 3, 4\}$ {child slot within each parent cell}
5: $C \leftarrow \text{ScatterToSlots}(X^{(\ell)}, \pi, \sigma)$ {$C \in \mathbb{R}^{|\mathcal{M}^{(\ell+1)}| \times 4 \times d}$}
6: $X^{(\ell+1)} \leftarrow \psi_\downarrow^{(\ell)}(C)$ {aggregate up to four children}
7: {Uncoarsen: expand parent channels and reshape into child slots, then scatter/gather to active children}
8: $U \leftarrow \text{reshape}(\text{MLP}^{(\ell)}(X^{(\ell+1)})) \in \mathbb{R}^{|\mathcal{M}^{(\ell+1)}| \times 4 \times d}$
9: $\widetilde{X}^{(\ell)} \leftarrow \text{GatherFromSlots}(U, \pi, \sigma)$ {scatter/gather back to active child sites}

---

$w$ along both axes to enable cross-window exchange. The partitioning induces a sparse *interaction list* of token pairs within each (shifted) window, and attention is computed only over these pairs via batched gather/scatter. In our implementation, we first build an integer index map as a lookup table, and then generate a rulebook (interaction list) from it using GPU tensor primitives. Attention is evaluated by batched gather/scatter along this rulebook, and the same rulebook can be shared across attention layers operating at the same scale. Relative positional bias terms are looked up by the relative offset $(\Delta i, \Delta j)$ within a window, matching Eq. (3).

### A.4. Coarsening and uncoarsening

The coarsening operator maps each active site $(i, j) \in \mathcal{M}^{(\ell)}$ to a parent index $(\lfloor i/2 \rfloor, \lfloor j/2 \rfloor) \in \mathcal{M}^{(\ell+1)}$ and groups up to four children per parent. For each parent, we concatenate its child features in a canonical order and apply a learned projection, as in Sec. 3.

The uncoarsening operator performs the inverse mapping by expanding each parent token into child tokens and scattering them to the corresponding active child sites. This produces the upsampled feature field $\widetilde{\mathcal{X}}^{(\ell)}$ used in the decoder before lateral fusion.

### A.5. Lateral fusion

At each decoder level, we concatenate the upsampled features with cached encoder features at the same lattice resolution and pass the fused tensor through the refinement block, as in Sec. 3.6.

### A.6. Coordinate augmentation and integer-grid mapping

During training, we apply coordinate-only augmentation to promote geometric robustness while keeping the patch encoder fixed when using precomputed patch embeddings. We randomly reflect, shear, and rotate the 2D coordinate set around its centroid, optionally drop a random subset of spots, translate the result so the minimum coordinate is at the origin, and map transformed coordinates to a collision-free integer grid. Concretely, we round coordinates to nearby lattice sites and resolve collisions by assigning the nearest available free lattice site, yielding integer indices suitable for sparse lattice operators.

### A.7. Default configuration used in experiments

Unless otherwise specified, experiments use the following non-trivial settings from the default configuration. We optimize **mean squared error (MSE)** in log-expression space and train for 100 epochs using AdamW (learning rate $10^{-4}$, weight decay $10^{-4}$) with cosine annealing to zero. We use **one slide per batch** and operate on precomputed patch embeddings by default. This enables efficient on-the-fly coordinate-only augmentations without re-encoding image patches. We apply coordinate augmentations with a cosine strength schedule that decays from 1.0 to 0.2 over training: mirror/shear/rotate are enabled with maximum shear angle $15°$ and rotation deviation up to $30°$ around right angles, and we apply spot dropout with base probability 0.2 (scaled by the same schedule). We use a single slide calibration token initialized as a learnable vector shared across slides, window half-size $w = 3$, attention dropout 0.05, and FFN expansion ratio 4. For patch embeddings of dimension 1024 (used by default), we project to model width $d = 512$ with 8 attention heads.

### A.8. Slide Calibration Token vs. Stain Normalization

A natural question is whether the slide calibration token simply re-derives information that classical color-space stain normalization already provides. To probe this, we compare three configurations on Brain (NCBI Geo): calibration token only (HiST default), classical Macenko stain normalization applied to image patches with the calibration token disabled, and both applied jointly. Table 5 reports the results. The two interventions are complementary: stain normalization alone is weaker than the learned calibration token, but combining them yields a further gain, suggesting that the calibration token captures slide-level signal beyond what color-space normalization can correct.

*Table 5.* Calibration token vs. Macenko stain normalization on Brain (NCBI Geo).

| Setting | $R^2 \uparrow$ | PCC $\uparrow$ |
|---|---|---|
| Stain normalization only (no $C_b$) | 0.20 | 0.51 |
| Calibration token only (default) | 0.23 | 0.54 |
| Both stain normalization and $C_b$ | **0.25** | **0.56** |

### A.9. Slide Calibration Token Analysis

To verify that the slide calibration token $C_b$ encodes slide-specific information rather than collapsing to a constant, we extract the final-layer $C_b$ representation of every slide within an organ cohort and analyze its discriminability across slides.

**Discriminability metrics.** We quantify discriminability of $C_b$ across slide groups on two cohorts (Kidney, Breast). For each cohort we report (i) 1-nearest-neighbor same-group accuracy on $C_b$, (ii) nearest-centroid accuracy (centroids computed per group), and (iii) the average within-group vs. between-group cosine distance. Results are summarized in Table 6. On both cohorts the same-group accuracies substantially exceed chance, and within-group cosine distances are smaller than between-group distances, indicating that $C_b$ encodes meaningful slide-level variation rather than collapsing to a constant.

*Table 6.* Discriminability of the slide calibration token $C_b$ across slide groups on two cohorts.

| Cohort | 1-NN same-group | Nearest-centroid acc | Within cos. dist | Between cos. dist |
|---|---|---|---|---|
| Kidney | 0.794 | 0.824 | 0.480 | 0.536 |
| Breast | 0.894 | 0.875 | 0.885 | 0.913 |

**PCA visualizations.** For two representative organ cohorts (Kidney and Breast), we project the per-slide $C_b$ onto its first two principal components in Figs. 4a and 4b. Slides cluster by source study rather than scattering uniformly, supporting the interpretation of $C_b$ as a learned slide-level summary that absorbs acquisition signatures.

## B. Theoretical Analysis

This appendix provides additional motivation and intuition for the hierarchical design of HiST. We present a multiresolution view of the proposed lattice operators, provide an informal receptive-field scaling comparison between fixed-resolution and hierarchical architectures, and discuss related approximation considerations.

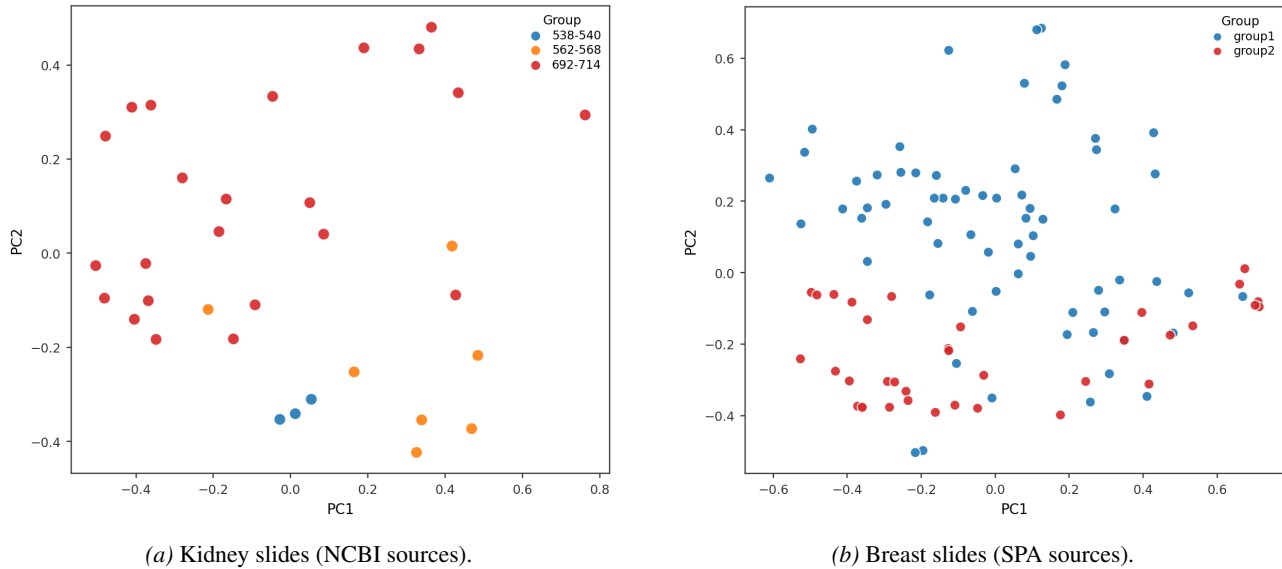

*(a)* Kidney slides (NCBI sources).          *(b)* Breast slides (SPA sources).

*Figure 4.* PCA of the slide calibration token $C_b$ across slides within an organ cohort. Slides cluster by source study rather than scattering uniformly, supporting the interpretation of $C_b$ as a learned slide-level summary that absorbs acquisition signatures.

## B.1. Multiresolution View via a Laplacian Pyramid

A classical way to formalize multiscale structure in 2D fields is to decompose a signal into a coarse component plus a sum of detail components across scales (e.g., Laplacian pyramids). For an idealized Laplacian pyramid, a spatial field $y$ admits a telescoping expansion:

$$y = \mathcal{U}_L \mathcal{A}_L y + \sum_{\ell=0}^{L-1} \Delta_\ell y, \tag{6}$$

where $\mathcal{A}_\ell$ is a dyadic averaging operator that coarsens resolution by a factor of $2^\ell$, $\mathcal{U}_L$ upsamples the coarsest field back to the original resolution, and $\Delta_\ell y$ represents detail coefficients at scale $\ell$.

We use this view as intuition for HiST's encoder–decoder design. Our dyadic coarsening operator plays the role of an analysis transform (approximating the coarse pathway), while the decoder's upsampling and lateral fusion re-inject same-scale details in the spirit of Laplacian residual reconstruction. The main point is not to claim that tissue expression is literally Laplacian-pyramid sparse, but that multiresolution operators provide a principled mechanism to rapidly aggregate context over expanding spatial neighborhoods while preserving fine-scale information.

**Concatenation vs. summation.** The idealized Eq. (6) reconstructs $y$ by *summing* the upsampled coarse field with detail coefficients at matched resolution. In HiST we instead *concatenate* the upsampled coarse features and the lateral encoder features along the channel dimension before passing them to a refinement block. This choice keeps both pathways available to the downstream attention layer, which can then learn data-dependent mixing weights, rather than committing the architecture to a fixed additive reconstruction. The pyramid is therefore an intuition for what information each pathway carries, not a literal description of the fusion algebra.

## B.2. Receptive Field Analysis

One of the primary challenges in whole-slide modeling is the *Deep Receptive Field Gap*: the discrepancy between the scale of local operators (patches 10s of $\mu$m) and the scale of biological organization (tissue 1cm).

**Receptive field radius (informal).** Let $\mathcal{O}$ be a composition of $L$ local operators, where each operator mixes information within a spatial radius $r$ on its input lattice. The receptive field radius $R_L$ is the maximum graph distance on the input lattice from which a node at layer $L$ can receive information.

**Fixed-resolution scaling (informal).** For a depth-$L$ network operating on a fixed lattice $\mathcal{L}_0$ with local kernel radius $r$ (e.g.,

$3 \times 3$ convolution or window attention), the receptive field grows at most linearly:

$$R_L^{\text{fixed}} = O(L \cdot r).$$

*Sketch.* Each local layer can expand support by at most $r$ lattice steps, so composing $L$ layers yields at most $Lr$ expansion (Luo et al., 2016).

**Implication.** To cover a spatial diameter $D$ (in lattice units), fixed-resolution designs generally require depth on the order of $D/r$ local-mixing layers.

**Hierarchical scaling (informal).** Now consider a hierarchical network with dyadic coarsening by a factor of 2 between stages. If stage $s$ operates on a lattice downsampled by $2^s$, then a local radius-$r$ layer at stage $s$ expands the receptive field by $O(r \cdot 2^s)$ on the input lattice. Summing across stages yields geometric growth; for example, with one local-mixing layer per stage,

$$R^{\text{hier}} \approx \sum_{s=0}^{S-1} r \, 2^s = r(2^S - 1),$$

up to constants and additional within-stage depth.

**Implication.** Hierarchy can reduce the number of stages needed to reach diameter $D$ from $O(D)$ to $O(\log D)$ (up to the chosen within-stage depth), enabling long-range context with relatively shallow networks.

### B.3. Multiscale Approximation and Spectral Bias

Why is hierarchy necessary if we only care about prediction? We appeal to signal processing principles.

**Heuristic (Spectral Decay of Biological Signals).** Biological spatial fields $y(\mathbf{s})$ (e.g., cell type maps, expression gradients) are comprised of smooth global trends $y_{\text{low}}$ and sparse local high-frequency details $y_{\text{high}}$. The energy of the signal generally decays with frequency.

**Intuition (Separation of Scales).** A hierarchical architecture can help separate the learning problem. Let $y = \sum_{\ell} y_{\ell}$ be a multiresolution expansion.

- Coarse layers operate on $\mathcal{L}_{\text{coarse}}$: They can efficiently approximate smooth functions $y_{\text{low}}$ because the Nyquist rate on the coarse lattice is sufficient for low-frequency signals.

- Fine layers operate on $\mathcal{L}_{\text{fine}}$: They only need to model the residual $y_{\text{high}} = y - y_{\text{low}}$. Since $y_{\text{high}}$ is typically sparse (edges, cell nuclei) or has lower variance, the effective complexity is reduced.

In contrast, a fixed-resolution network may need to propagate local mixing operations across the fine lattice to capture the same global trends, which can be less efficient than explicitly introducing multiresolution operators.

### B.4. Slide Calibration Token as Domain Alignment

We provide a simple probabilistic interpretation of the slide calibration token mechanism $\text{MCA}(X, C_b)$.

Let $P(\mathbf{y}|\mathbf{x}, z)$ be the true generative process, where $z$ is a slide-specific latent variable (e.g., staining intensity). Standard regression estimates $E[\mathbf{y}|\mathbf{x}]$, marginalized over $z$. If $z$ varies strongly across slides (batch effect), the marginal prediction is high-variance. Our slide calibration token $C_b$ can be viewed as an estimator $\hat{z} = \phi(\{\mathbf{x}_n\}_{n=1}^N)$ of this latent nuisance factor.

**Mechanism.** With a single calibration token ($C_b = \mathbf{c}_b \in \mathbb{R}^{1 \times d}$), the update $X' = X + \text{Attn}(X, C_b, C_b)$ broadcasts a slide-level correction:

$$\mathbf{x}_i' = \mathbf{x}_i + (\mathbf{c}_b W_V).$$

If $C_b$ encodes the shift $\Delta\mu_{slide}$, this additive term acts as a learned, slide-specific bias that aligns tissue features to a canonical domain. Restricting $C_b$ to a single token enforces a low-bandwidth global bottleneck, encouraging the model to capture *global* properties of the slide distribution rather than memorizing local artifacts.

## C. Additional Qualitative Results

This appendix gathers additional qualitative visualizations complementing the quantitative results.

## C.1. Spatial Visualizations

We visualize predicted vs. measured spatial expression heatmaps for canonical marker genes across four organ contexts. For each gene, the panel shows the H&E image alongside measured and HiST-predicted log-expression maps at the measured locations. HiST recovers the broad spatial organization of these markers and preserves locally coherent structure.

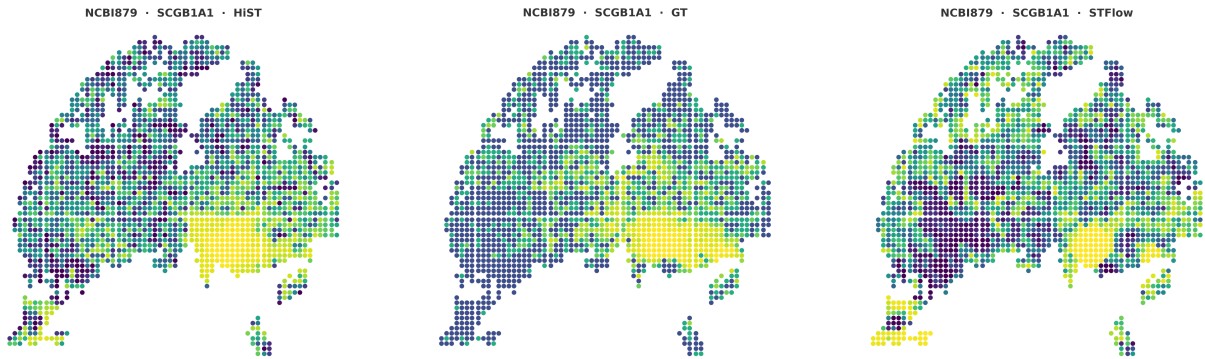

*Figure 5.* Lung slide NCBI879, marker gene *SCGB1A1*.

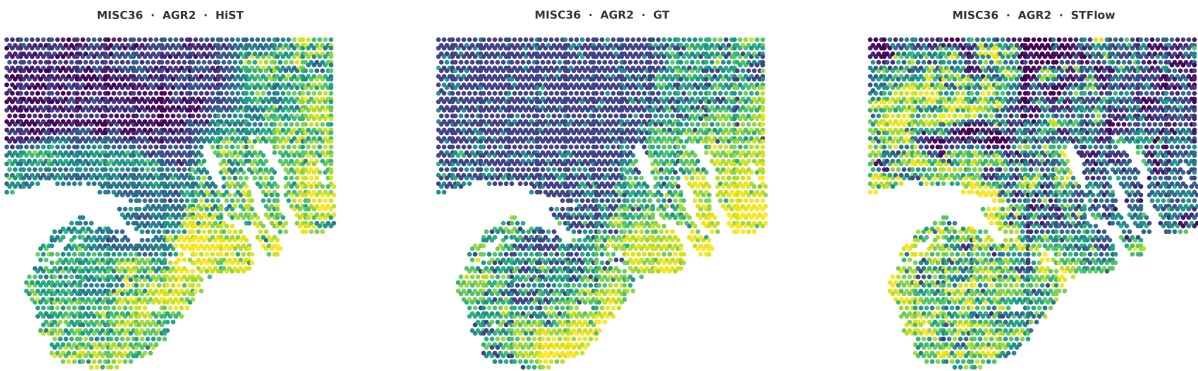

*Figure 6.* Colon slide MISC36, marker gene *AGR2*.

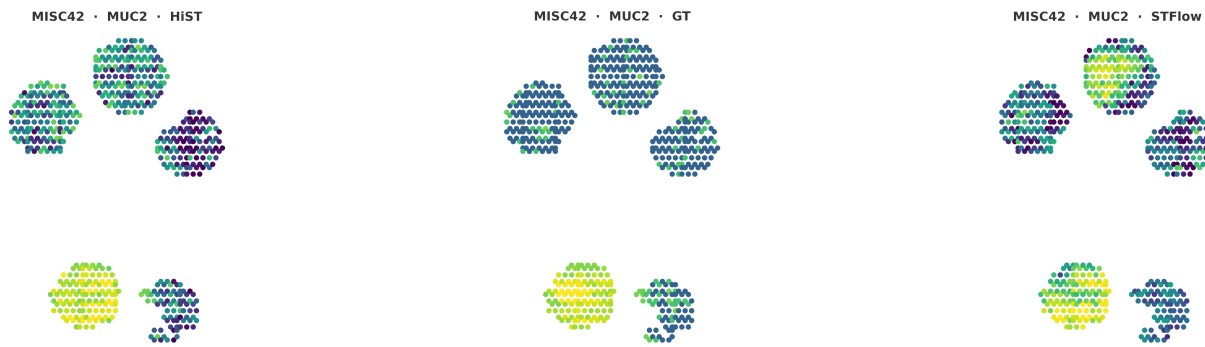

*Figure 7.* Colon slide MISC42, marker gene *MUC2*.

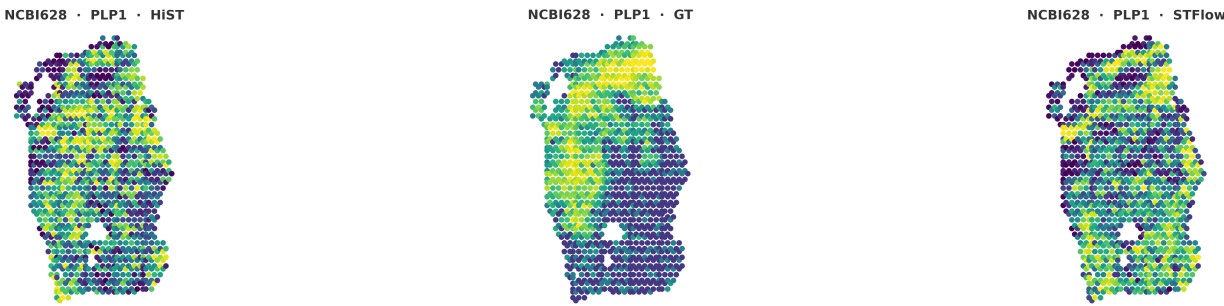

*Figure 8.* Brain slide NCBI628, marker gene *PLP1*.

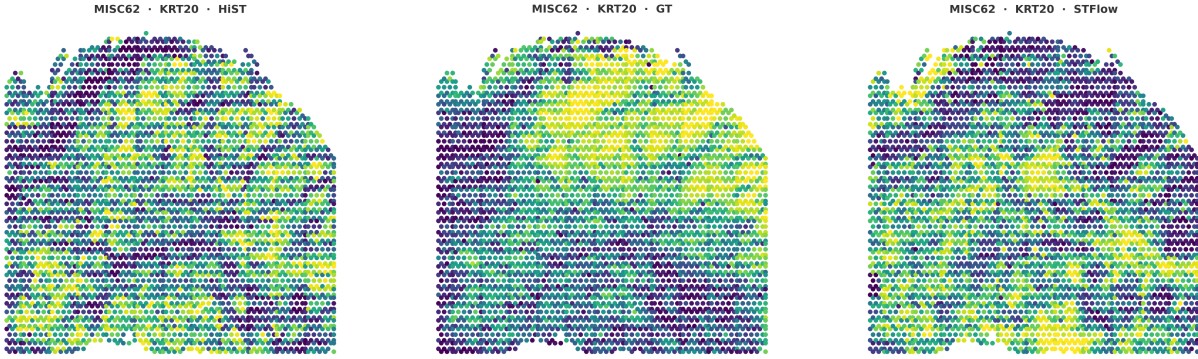

*Figure 9.* Colon slide MISC62, marker gene *KRT20*.

