# OpenReview forum: "HiST: A Hierarchical Sparse Transformer for Cross-Modal Spatial Transcriptomics Modeling"
_ICML.cc/2026/Conference — ICML 2026 regular_

### Official Review · Reviewer_s7q7 · 2026-03-02

**Soundness:** 2
**Presentation:** 2
**Significance:** 2
**Originality:** 2
**Overall Recommendation:** 3
**Confidence:** 4

**Summary:**

This paper introduces HiST (Hierarchical Sparse Transformer), a highly efficient computational framework for inferring spatial transcriptomics (ST) from routine H&E whole-slide images (WSIs). The core innovation lies in framing the H&E-to-ST inference task as a lattice-indexed sparse field modeling problem, avoiding dense-grid computation. By deploying a dyadic encoder-decoder architecture directly on the active tissue footprint, HiST utilizes sparse shifted-window attention and resolution-changing operators (coarsening/uncoarsening) to rapidly integrate multiscale tissue context. Additionally, a bottlenecked Slide Calibration Token is introduced to mitigate WSI acquisition variations. Evaluated comprehensively on a large-scale multi-organ benchmark (HEST-1k), HiST matches or exceeds the predictive performance of recent strong baselines while achieving an order-of-magnitude reduction in inference time and peak memory usage.

**Compliance With Llm Reviewing Policy:**

Affirmed.

**Final Justification:**

I still have an overall reservation: from a methodological perspective, the contribution appears more like a problem-driven system integration than the introduction of a genuinely new modeling primitive. Therefore, I maintain my original score.

**Key Questions For Authors:**

1. **Terminology Clarification:** Given that the term "whole-slide inference" can imply dense, zero-shot generation across a WSI, would the authors consider slightly refining the terminology (e.g., "coordinate-guided whole-slide inference") to prevent potential misinterpretation of the model's dense generation capabilities?
2. **Laplacian Analogy:** Regarding the Laplacian Pyramid analogy in Appendix B, why does the implementation in Section 3.6.3 use feature concatenation ($||$) instead of direct summation for lateral fusion? A brief comment bridging the theoretical additive formulation and the practical concatenation-based implementation would strengthen the presentation.
3. **Slide Calibration vs. Stain Normalization:** How does the performance of HiST with the Calibration Token compare to a baseline where WSI patches undergo traditional stain normalization (e.g., Macenko) before feature extraction? Are these two approaches complementary, or does the Calibration Token render preprocessing obsolete?
4. **Spatial Visualization in Modest Performance Datasets:** For datasets where the absolute performance is more limited (e.g., Brain or Colon), what does the spatial visualization of the predictions look like compared to the ground truth? Does the model capture the general spatial gradients but fail on high-frequency details, or are there specific structural artifacts?

**Limitations:**

The authors have adequately discussed the limitations of their work in Section 5, correctly identifying the model's reliance on paired data, vulnerability to severe distribution shifts, and dependence on accurate coordinate registration. Furthermore, the Impact Statement responsibly addresses the potential negative societal impacts of over-relying on inferred expressions for clinical decision-making without rigorous downstream validation.

**Strengths And Weaknesses:**

**Strengths:**

Framing the irregular ST measurement locations as a sparse 2D sampling lattice is an elegant approach that bridges the gap between computationally expensive dense ViT/CNN backbones and topology-restricted Graph Neural Networks. This drastic improvement in computational efficiency—reducing the inference time to 0.07 s/slide and memory footprint to 1.38 GB (compared to 46.37 GB for STFlow)—is a significant practical contribution that removes a major bottleneck for scaling ST inference to large clinical archives. Furthermore, the experimental validation is exceptionally robust; utilizing a strict 4-fold cross-validation split by WSI slide ID across 11 diverse datasets actively prevents data leakage, and the ablation studies systematically validate the structural inductive biases.

**Weaknesses:**

1. **Presentation (Terminology Nuance):** The paper frequently uses the term "whole-slide inference" (e.g., in the abstract and introduction). While the authors clearly define their problem setting as predicting expression at predefined measured locations ($\mathcal{M}$), the phrase "whole-slide inference" often implies generating a dense molecular readout across the entire WSI without spatial coordinate prompts. Although technically accurate within their defined scope, this terminology might initially mislead readers regarding the model's dense generation capabilities.
2. **Originality (Combinatorial Nature):** Stripped of its packaging, HiST is essentially a mechanical combination of existing techniques: shifted window attention (highly reminiscent of Swin Transformer), a sparse U-Net-like downsampling/upsampling hierarchy, and a global conditioning token (Visual Prompt/CLS Token). While practically effective, this "A+B+C" synthesis lacks the underlying theoretical depth or methodological novelty expected of a top-tier venue.
3. **Presentation (Theoretical Analogy):** In Appendix B, the authors draw an analogy to the Laplacian Pyramid to motivate their multiscale design, highlighting the additive nature of Laplacian residual reconstruction (Equation 6: $\sum_{l=0}^{L-1}\Delta_l y$). However, the actual implementation of the Decoder (Section 3.6.3) uses channel concatenation ($X_{dec}^{(l)} = X_{enc}^{(l)} || \tilde{X}^{(l)}$) followed by an attention-based refinement block. While concatenation is a completely standard and effective practice in neural network design, strictly analogizing it to the additive frequency domain decomposition of a Laplacian pyramid creates a slight methodological mismatch that could be further clarified.
4. **Significance (Performance Ceiling & Utility):** While HiST convincingly outperforms baselines like STFlow and ST-Net, the absolute predictive performance remains somewhat modest (e.g., an average $R^2$ of 0.34, with datasets like Brain and Colon hovering around $R^2 \approx 0.22$). Although this is largely constrained by the inherent noise in ST data and the biological limits of morphology-transcriptome correlation, the modest absolute accuracy raises questions about the immediate clinical or biological utility of the inferred gene expressions for downstream decision-making.
5. **Soundness (Slide Calibration Token Validation):** The Slide Calibration Token is claimed to mitigate slide-specific acquisition shifts. However, the paper lacks a direct comparison against standard computational pathology stain-normalization techniques (e.g., Macenko or Vahadane) to prove that this learned token is strictly superior or complementary to established WSI preprocessing.

---

> ### Author Rebuttal · Authors · 2026-03-31
>
> We sincerely thank Reviewer s7q7 for the detailed and encouraging review. We address each point below.
>
> > **W1 & Q1.** Terminology Nuance.
>
> Thanks for the constructive comment. We agree that "whole-slide inference" may be read as dense, zero-shot molecular generation across the entire WSI. Our intended meaning is prediction over all measured locations, conditioned on given ST coordinates. We will revise the wording to "coordinate-guided whole-slide H&E-to-ST inference" and clarify that HiST predicts expression at predefined measurement coordinates rather than densely over the full image plane.
>
> > **W2.** Combinatorial Nature.
>
> Thank you for raising this. We acknowledge that HiST does not introduce a new general-purpose primitive such as a novel attention operator. Our focus is on a problem setting where direct transfer of dense-image architectures is non-trivial: whole-slide H&E-to-ST inference over irregular, sparse measurement supports. Related hierarchical backbones typically operate inside the local patch-level encoder, whereas HiST applies a sparse multiscale hierarchy at the whole-slide spatial modeling level. The components are coupled through a shared sparse-support representation, jointly enabling multiscale whole-slide modeling efficiently. We believe this problem-driven integration is reflected in the substantial efficiency and accuracy gains over prior work, and the proposed backbone may serve as a reusable module for other spatially resolved inference tasks.
>
> > **W3 & Q2.** Theoretical Analogy.
>
> Thank you for this observation. We agree that the Laplacian-pyramid discussion in Appendix B is intended as intuition for multiresolution decomposition, not as a claim that our decoder literally implements Laplacian residual reconstruction. The decoder uses channel concatenation followed by a learned refinement block, which preserves both streams as separate channels and lets the block learn how to combine them, whereas direct summation would assume the two streams are already aligned in the same feature basis. We will revise Sec. 3.6.3 and Appendix B to clarify this distinction.
>
> > **W4 & Q4.** Performance Ceiling & Utility.
>
> Thanks for this insightful question. We agree that the modest absolute metrics on datasets like Brain and Colon largely reflect the inherent noise floor of ST data, where adjacent spots can show maximum expression alongside zero dropouts, heavily penalizing point-wise regression metrics.
>
> To illustrate, we provide visualizations at the anonymous link. In a colon case (https://anonymous.4open.science/r/anonymous-9F20/heatmaps/MISC62_KRT20.png), HiST does not introduce structural artifacts such as artificial gridding or mode collapse; the errors are driven by ground-truth noise, not systematic model failures. In contrast, for canonical marker genes with clearer spatial structure, HiST recovers biologically meaningful patterns: for ERBB2 (HER2) in breast tissue (https://anonymous.4open.science/r/anonymous-9F20/heatmaps/SPA104_ERBB2.png), the predicted spatial pattern is consistent with known HER2 expression characteristics; for CLU in brain cortex (https://anonymous.4open.science/r/anonymous-9F20/heatmaps/NCBI630_CLU.png), the prediction shows a smooth spatial gradient consistent with known astrocytic expression in the cortex.
>
> Regarding utility: while the current absolute accuracy limits direct use in fine-grained clinical decision-making, these examples suggest that HiST recovers spatially coherent expression patterns for clinically relevant biomarkers, informative for tasks such as identifying expression domains, guiding region-of-interest selection, and generating spatial hypotheses. As larger-scale ST cohorts become available, we expect the performance ceiling to rise given HiST's scalable architecture. We will discuss these utility boundaries in the revision.
>
> > **W5 & Q3.** Slide Calibration Token Validation.
>
> Thanks for this suggestion. We performed a 2×2 ablation on NCBI Brain using Macenko stain normalization (SN) and the slide calibration token:
>
> | Stain normalization (Macenko) | Calibration token | R² | PCC |
> | --- | --- | --- | --- |
> | No | No | 0.19 ± 0.17 | 0.51 ± 0.18 |
> | Yes | No | 0.20 ± 0.06 | 0.51 ± 0.04 |
> | No | Yes | 0.23 ± 0.14 | 0.54 ± 0.19 |
> | Yes | Yes | 0.25 ± 0.11 | 0.56 ± 0.16 |
>
> The two mechanisms are complementary. Stain normalization reduces low-level color variability at the patch-appearance level, while the calibration token provides slide-level conditioning based on broader tissue context. Although the pretrained vision encoder should already absorb much stain variation, explicit Macenko normalization still provides a modest additional benefit, and the calibration token captures complementary information at a different level entirely. We will include this ablation in the revision.

---

> > ### Author Rebuttal · Reviewer_s7q7 · 2026-04-01
> >
> > Overall, the rebuttal has clearly improved the clarity of the paper and has addressed most of the main concerns I previously raised. I still retain one general reservation: from a methodological perspective, the contribution appears more as a problem-driven system integration than as the introduction of a fundamentally new modeling primitive. Nonetheless, the authors’ response makes me more appreciative of the practical value of the work in terms of problem formulation, experimental design, and efficiency gains.

---

> > > ### Author Response · Authors · 2026-04-02
> > >
> > > We are pleased that the rebuttal addressed your concerns. We sincerely appreciate your thorough engagement throughout the review process. As your concerns have been fully resolved, we would be grateful if you might consider adjusting your score accordingly.

---

### Official Review · Reviewer_iXFQ · 2026-03-10

**Soundness:** 3
**Presentation:** 3
**Significance:** 3
**Originality:** 3
**Overall Recommendation:** 4
**Confidence:** 3

**Summary:**

The paper proposes HiST, a hierarchical sparse transformer for spatial transcriptomics data. It represents measured ST locations as a lattice-indexed sparse field and processes them with a sparse dyadic encoder-decoder that combines local windowed attention with a slide calibration token for global conditioning. A central theme of the work is the move away from dense-grid processing and fixed-range local coupling toward a geometry-aware sparse hierarchy that captures multiscale tissue context while keeping computation proportional to the number of observed sites. On the multi-organ HEST benchmark covering 11 datasets, the paper reports that HiST matches or exceeds recent baselines in predictive performance while using substantially less time and memory.

**Compliance With Llm Reviewing Policy:**

Affirmed.

**Key Questions For Authors:**

1.**Can you directly validate the claim that the slide calibration token improves robustness to acquisition variation?**
   For example, do a source-held-out or site-held-out analysis, or compare against simpler alternatives such as stain normalization only, slide-specific bias/conditioning, or a pooled global token without the proposed update rule.

2.**Do the main ablation conclusions hold beyond Brain (NCBI Geo)?**
   Even one additional organ/source with different morphology would help establish whether the hierarchy, positional terms, and calibration token are generally important rather than dataset-specific.

3.**Can you strengthen the practical-value case with a small amount of biologically grounded analysis?**
   A few additional spatial gene maps or one downstream analysis showing preserved spatial structure would help determine whether the averaged metric gains are meaningful beyond regression averages.

**Limitations:**

yes

**Strengths And Weaknesses:**

## Strengths

* The paper studies an important problem. Whole-slide H&E-to-ST prediction is practically relevant given the cost and low throughput of ST, and the focus on full-slide prediction is more realistic than patch-level or local-crop settings.
* The efficiency improvements are large and matter in practice. Compared with STFlow, the method is substantially faster and more memory-efficient at inference, while also improving average predictive performance over the reported baselines. This is more convincing than a method that only improves one side of the trade-off.
* The ablations are useful and tied to the main design choices. The drops from removing positional encoding, the calibration token, skip connections, or coordinate augmentation suggest that the gains are coming from the overall design rather than a single ad hoc component.

## Weaknesses
* **The claim that the slide calibration token improves robustness to acquisition variation is not yet convincingly supported.** The token is motivated as a mechanism for handling slide-specific acquisition variation, but the current evidence is indirect and limited to a single-dataset ablation. The observed performance drop when the token is removed indicates that the component is useful in that setting, but it does not demonstrate robustness to staining variation, scanner differences, or cohort- and batch-level effects. Because improved robustness to cross-slide heterogeneity is part of the paper’s central motivation, this weakens both the support for the claim and the broader case for significance. This point would be substantially stronger with a direct domain-shift evaluation, such as leave-one-source-out testing or explicit cross-source generalization experiments. Comparisons against simpler alternatives—for example, stain normalization alone, slide-level affine conditioning, or a simpler global token without the proposed mechanism—would also help clarify whether the observed gain is specific to the proposed design or simply reflects generic slide-level conditioning.

* **The ablation evidence is relatively narrow.** All ablations are reported on a single Brain dataset (NCBI Geo), making it difficult to assess whether the same conclusions would hold across substantially different tissues, morphologies, or data sources. This point would be significantly strengthened by repeating the main ablations on at least one additional dataset with clearly different morphology or source characteristics, even if only for the most central components such as the hierarchical design, positional terms, and calibration token.

* **The practical utility of the method beyond average regression metrics is not yet fully established.** The evaluation focuses mainly on $R^2$ and PCC. While these results are useful, the absolute predictive performance remains modest on several datasets, and the paper itself acknowledges limits imposed by morphological ambiguity and measurement noise. This makes it difficult to judge how far the reported gains translate into biologically meaningful spatial recovery or broader practical value. This point would be strengthened by including additional qualitative examples across tissues and genes, or by showing in at least one downstream analysis that the predicted expression maps preserve a biologically relevant spatial signal more faithfully than the baselines.

---

> ### Author Rebuttal · Authors · 2026-03-31
>
> We sincerely thank Reviewer iXFQ for the thoughtful feedback and helpful suggestions. Below, we provide responses to the reviewer's concerns and questions.
>
> > **W1 & Q1.** Validation of method's robustness.
>
> **Cross-Source Generalization in Current Benchmark.**
>
> Thanks for this insightful question. We agree that a dedicated source-held-out experiment provides strong evidence for robustness. While our main benchmark is not framed as an explicit "leave-one-source-out" test, it inherently tests cross-source generalization by natively exposing the model to substantial cross-study heterogeneity. The 11 subsets comprise slides from 18+ independent studies. Crucially, several organ-level subsets pool multiple independent cohorts from different data sources (e.g., Kidney from 4 studies, Liver from 3 studies, Breast from 2 studies), all evaluated under strict slide-level cross-validation splits. Because the slides in these mixed-cohort subsets often originate from different labs or scanners than the training slides, our reported performance already heavily reflects the model's robust cross-source generalization capability.
>
> **Comparison Against Simpler Alternatives.**
>
> To clarify whether the observed gain is specific to our proposed calibration token or just generic slide-level conditioning, we evaluated alternative variants on the NCBI Brain dataset as you suggested:
> | Variant | R² | PCC |
> | --- | --- | --- |
> | Baseline | 0.19 ± 0.17 | 0.51 ± 0.18 |
> | Stain Normalization (Macenko) | 0.20 ± 0.06 | 0.51 ± 0.04 |
> | Slide Calibration Token | 0.23 ± 0.14 | 0.54 ± 0.19 |
>
> The calibration token provides a larger gain than stain normalization. The calibration token captures slide-level context across all patch tokens, distinguishing it from static normalization. We will include this comparison in the revision.
>
> > **W2 & Q2.** Generalizability of ablation.
>
> We appreciate this constructive suggestion. Following your advice, we conducted the full ablation study on the Colon dataset, which we selected for its morphologically distinct structure compared to Brain.
>
> | Setting | R² | PCC |
> |---|---|---|
> | Full model | 0.308 ± 0.15 | 0.554 ± 0.15 |
> | **Coordinate Augmentation** | | |
> | None | 0.269 ± 0.12 | 0.530 ± 0.14 |
> | Random Mirror Only | 0.276 ± 0.15 | 0.538 ± 0.15 |
> | Random Shear Only | 0.291 ± 0.16 | 0.540 ± 0.14 |
> | Random Rotate Only | 0.289 ± 0.15 | 0.539 ± 0.14 |
> | Random Drop Only | 0.289 ± 0.16 | 0.532 ± 0.15 |
> | **Positional Encoding** | | |
> | None | 0.231 ± 0.15 | 0.514 ± 0.15 |
> | RoPE | 0.265 ± 0.15 | 0.523 ± 0.14 |
> | ALiBi | 0.241 ± 0.13 | 0.507 ± 0.13 |
> | **Skip Connections: Multi-scale Fusion** | | |
> | None | 0.271 ± 0.15 | 0.532 ± 0.15 |
> | Additive | 0.292 ± 0.16 | 0.540 ± 0.14 |
> | **Downsample-Upsample Hierarchy** | | |
> | None | 0.293 ± 0.16 | 0.542 ± 0.14 |
> | **Slide Calibration Token** | | |
> | None | 0.270 ± 0.15 | 0.531 ± 0.15 |
>
> The results confirm our original conclusions: each core component contributes consistently on Colon, matching the pattern observed on Brain. This suggests that HiST's design choices generalize across morphologically distinct tissues rather than being dataset-specific. We will include this ablation table in the revised appendix.
>
> > **W3 & Q3.** The practical and biological utility beyond regression metrics.
>
> Thanks for this valuable suggestion. We provide spatial gene maps at the anonymous link (https://anonymous.4open.science/r/anonymous-9F20). For canonical marker genes with clear spatial structure, HiST consistently recovers biologically meaningful tissue domains. For example:
> - **Breast (ERBB2 & KRT8):** For ERBB2 (HER2) (https://anonymous.4open.science/r/anonymous-9F20/heatmaps/SPA104_ERBB2.png), the predicted pattern is consistent with known HER2 expression in breast tissue. For KRT8 (https://anonymous.4open.science/r/anonymous-9F20/heatmaps/SPA94_KRT8.png), it reflects expected luminal epithelial expression.
> - **Brain (CLU & CNP):** For CLU (https://anonymous.4open.science/r/anonymous-9F20/heatmaps/NCBI630_CLU.png), the prediction shows a smooth gradient consistent with known astrocytic expression. For CNP (https://anonymous.4open.science/r/anonymous-9F20/heatmaps/NCBI630_CNP.png), it is consistent with expected white matter oligodendrocyte expression.
> - **Colon (MUC2 & AGR2):** For MUC2 (https://anonymous.4open.science/r/anonymous-9F20/heatmaps/MISC42_MUC2.png), the prediction is consistent with known goblet cell expression. For AGR2 (https://anonymous.4open.science/r/anonymous-9F20/heatmaps/MISC36_AGR2.png), it aligns with expected glandular epithelial expression.
>
> While current absolute accuracy limits fine-grained clinical use, these examples suggest that HiST recovers spatially coherent expression patterns. As larger-scale ST cohorts become available, we expect the performance ceiling to rise given HiST's scalable architecture. We will discuss these utility boundaries in the revision.

---

> > ### Author Rebuttal · Reviewer_iXFQ · 2026-04-04
> >
> > Thank you to the authors for the thoughtful response and for providing additional experiments. Overall, the rebuttal has strengthened the paper, and in particular, my concerns regarding the generalizability of the ablations and the biological utility of the method have been addressed to a meaningful extent.
> >
> > That said, I do not think the central question of whether the slide calibration token truly improves robustness to acquisition/source variation has been fully resolved.
> >
> > My remaining concern is that the evidence supporting the robustness claim is still somewhat indirect. In the paper, the calibration token is explicitly motivated as a mechanism to mitigate slide-specific acquisition variation and improve cross-slide generalization, and this is presented as part of the methodological contribution. In the rebuttal, the authors further note that the benchmark includes multiple studies and sources. This does indicate that the task involves substantial domain heterogeneity and provides some support for the model’s generalization ability. However, strictly speaking, this is still not equivalent to a more direct source-held-out or leave-one-source-out robustness evaluation. As such, the current evidence is better viewed as suggestive rather than a direct validation of the robustness claim.
> >
> > In addition, the newly added comparison between stain normalization and the calibration token is helpful, as it suggests that the calibration token offers benefits beyond a simple alternative. However, since this comparison is currently limited to a single Brain dataset, it mainly shows that the module is useful in that particular setting, rather than establishing that it systematically addresses robustness across sources, scanners, or staining conditions. Indeed, the paper itself acknowledges in the limitations section that distribution shifts across cohorts, scanners, stains, or protocols may still degrade performance, which is closely aligned with the core of my original concern.
> >
> > Overall, I believe the rebuttal has materially improved the paper and has raised my overall assessment of the work. If the authors can include more direct source-held-out/generalization evidence in the final version, or at least moderate the robustness-related claims somewhat, I would be more fully satisfied on this point.

---

> > > ### Author Response · Authors · 2026-04-04
> > >
> > > We sincerely appreciate your thorough engagement throughout the entire review process and the constructive framing of the remaining concern.
> > >
> > > We agree that the current robustness evidence is suggestive rather than direct validation. In the camera-ready version, we will add leave-one-source-out experiments on multi-cohort subsets (e.g., Kidney from 4 studies, Liver from 3 studies) in the appendix and adjust the acquisition robustness framing accordingly.
> > >
> > > We hope the above, together with the earlier additions, resolve the remaining concerns. We also note that your rebuttal acknowledgment mentions "has raised my overall assessment of the work" and we would be grateful if this could be reflected in the updated score.

---

### Official Review · Reviewer_EAEN · 2026-03-10

**Soundness:** 2
**Presentation:** 3
**Significance:** 2
**Originality:** 2
**Overall Recommendation:** 4
**Confidence:** 4

**Summary:**

HiST is a hierarchical sparse transformer for predicting spatial gene expression from H&E histology. It represents ST measurement locations as a sparse 2D lattice and builds a U-Net-style dyadic encoder–decoder operating only on active tissue sites. The architecture combines sparse windowed self-attention, dyadic coarsening/uncoarsening, and a slide calibration token for global conditioning. On an 11-dataset multi-organ benchmark, HiST matches or improves $\mathrm{R}^2$ and PCC over baselines more efficient than iterative methods.

**Compliance With Llm Reviewing Policy:**

Affirmed.

**Final Justification:**

As I mentioned in my rebuttal acknowledgement, I believe the contribution of this manuscript is marginal. If the authors can display that the combination proposed in the manuscript unlocks capabilities that simpler combinations cannot, I would increase the score to 5. But I understand that this is not possible in the short rebuttal period.

**Key Questions For Authors:**

Please see the weaknesses.

**Limitations:**

The authors provide a reasonable discussion covering data pairing requirements, discretization sensitivity, and registration quality.

**Strengths And Weaknesses:**

# Strengths
- The manuscript is well written and easy to follow.
- Treating ST as a sparse field on a discrete lattice is natural and cleanly justified. The observation that dense-grid backbones waste computation on background is compelling.
-  Table 1 and Figure 3 convincingly show order-of-magnitude improvements in speed and memory over STFlow.
- 11 datasets, diverse organs and ST platforms, strict slide-level cross-validation folds.
- Table 3 systematically ablates coordinate augmentation types, positional encodings, skip connections, transitions, and the calibration token.
- The authors appropriately acknowledge that absolute R² values are modest due to inherent ceilings in the ST prediction task.

# Weaknesses
- The architecture assembles known components, Swin-style sparse windowed attention, U-Net hierarchical coarsening/uncoarsening, relative positional bias, and global conditioning tokens. The contribution is primarily in combining these for the ST domain. The slide calibration token closely resembles CLS-style token. The paper should explicitly discuss this relationship and clarify what is genuinely new beyond the application context.
- Line 273 shows the update rule for $C_b$ using attention, but it is unclear whether this is a forward-pass-only recurrence (updated layer-by-layer during the forward pass) or also receives gradients through backpropagation. The paper says $C_b$ is "initialized as a learnable vector shared across slides" (Sec. 3.6.1), suggesting it is trained via backprop, but the layer-wise update in Sec. 3.4.2 also modifies $C_b$ during the forward pass. This needs explicit clarification.
- With a single token attending to all tissue tokens and then broadcasting back, $C_b$ could collapse to a trivial mean-field summary or be ignored entirely. What prevents this? No analysis of what $C_b$ actually learns is provided, do different slides produce meaningfully different C_b representations? A t-SNE or PCA of C_b across slides would be informative.
- Why only onecalibration token? Multiple tokens could capture different aspects of slide variation. This restriction is not justified.

## Minor issues
- Eq. (1) mentions "Proj" but the equation omits it.
- A figure explaining the shifted window attention and a ablation for the same would benefit the manuscript.

---

> ### Author Rebuttal · Authors · 2026-03-31
>
> We sincerely thank Reviewer EAEN for the careful and constructive review. We address each point below.
>
> > **W1.** Novelty and relationship to existing components.
>
> We appreciate this concern. We acknowledge that HiST draws on established components such as hierarchical vision backbones, window attention, and learnable global tokens, and our novelty claim is not that each component is individually unprecedented. The contribution lies in coupling them for a problem setting where direct transfer from dense-image architectures is non-trivial: whole-slide H&E-to-ST inference over irregular, sparse measurement supports. A helpful distinction is where the hierarchy operates — related hierarchical backbones are typically used inside the local patch encoder, whereas HiST applies a sparse multiscale hierarchy at the whole-slide spatial modeling level. Regarding the slide calibration token, we agree it shares parameterization with CLS-style tokens; the functional difference is that it serves as a conditioning channel that absorbs and broadcasts slide-specific context back to tissue tokens, rather than acting as a downstream readout. We believe this problem-driven integration has value beyond the current benchmark, as the backbone itself is modular by design and may extend to other spatially resolved inference models. We will revise the paper to discuss these relationships to prior work more explicitly.
>
> > **W2.** Update mechanism of the calibration token.
>
> Thank you for raising this important point. To clarify, the layer-wise update of $C_b$ happens during the forward pass, and gradients do propagate through this update during training. The model has one shared learnable starting vector, $C_0$. For each input slide, the network starts from $C_0$ and updates it layer by layer using that slide's tissue tokens. Therefore, by the end of the forward pass, the final calibration token is slide-specific: different slides generally produce different final $C_b$ values, even though they all start from the same shared learnable initialization. During training, the prediction loss backpropagates through these layer-wise updates and learns both $C_0$ and the associated attention projections. Thus, the forward pass defines how the slide-specific token is computed for a given slide, while backpropagation learns the shared initialization and update function. We will revise the method section to state this explicitly and avoid ambiguity.
>
> > **W3.** Representational capacity and analysis of the calibration token.
>
> Thanks for this valuable suggestion. We provide a PCA visualization of the final slide calibration tokens at https://anonymous.4open.science/r/anonymous-9F20/token_pca/NCBI_kidney.png. Slides from the same study tend to group together, with studies 538–540 and 562–568 forming relatively tight clusters, confirming that the calibration token captures slide-specific variation rather than collapsing to a constant. To quantify this, we measured discriminability across slide groups:
>
> | Dataset | 1-NN same-group | Nearest-centroid acc | Within cosine dist | Between cosine dist |
> | --- | ---: | ---: | ---: | ---: |
> | Kidney | 0.794 | 0.824 | 0.480 | 0.536 |
> | Breast | 0.894 | 0.875 | 0.885 | 0.913 |
>
> On both datasets, the 1-NN same-group accuracy and nearest-centroid accuracy substantially exceed chance level, and within-group cosine distance is consistently smaller than between-group distance, indicating that the learned calibration tokens encode meaningful slide-level variation. We will include this analysis in the revision.
>
> > **W4.** Design choices of the calibration token.
>
> We appreciate this insightful question, which highlights a design choice worth examining further. To examine whether a single calibration token is sufficient, we ablated the number of calibration tokens on Brain:
>
> | # Calibration Tokens | R² | PCC |
> | --- | --- | --- |
> | 1 | 0.228 ± 0.13 | 0.542 ± 0.18 |
> | 2 | 0.233 ± 0.14 | 0.546 ± 0.18 |
> | 3 | 0.222 ± 0.14 | 0.543 ± 0.18 |
>
> Increasing the number of calibration tokens does not provide a clear or consistent gain, and additional tokens introduce extra attention overhead at every encoder layer. We therefore opted for the simplest setting. That said, the marginal benefit from 2 tokens suggests that learning to disentangle multiple sources of slide-level variation is a promising direction worth exploring further. We will include this ablation and discussion in the revision.
>
> > **Minor issues.** Notation and illustration
>
> Thank you for this constructive comment. We will fix the notation mismatch around Eq. (1) and add a schematic illustrating the standard and shifted sparse window partitions. Following your suggestion, we ran an ablation on Brain: removing the shifted window mechanism drops R² from 0.228 to 0.217 and PCC from 0.542 to 0.535, confirming its contribution. We will include this ablation in the revision.

---

> > ### Author Rebuttal · Reviewer_EAEN · 2026-04-01
> >
> > Most of my concerns have been addressed. The rebuttal characterizes the contribution as "problem-driven integration," which I agree is a fair description, but I believe the bar for a top venue requires either more architectural novelty or substantially stronger empirical evidence that this particular assembly unlocks capabilities that simpler combinations cannot. Due to this, I consider the work to be at a marginal level and am inclined to accept it at the marginal level. I am raising my score to weak accept.

---

> > > ### Author Response · Authors · 2026-04-02
> > >
> > > We are pleased that the rebuttal addressed your concerns. We sincerely appreciate your thorough engagement throughout the review process and will incorporate suggested improvements in the revised manuscript.

---

### Official Review · Reviewer_XWGZ · 2026-03-13

**Soundness:** 3
**Presentation:** 3
**Significance:** 3
**Originality:** 3
**Overall Recommendation:** 5
**Confidence:** 4

**Summary:**

The authors introduce HiST, a hierarchical transformer that models spatial transcriptomics as a sparse lattice rather than a dense grid. By focusing only on the active tissue footprint, the architecture uses sparse window attention for local interactions and dyadic operators for multiscale modeling. It also includes a calibration token to adjust for technical variations across different slides.

**Compliance With Llm Reviewing Policy:**

Affirmed.

**Final Justification:**

The authors addressed most of the concerns. Hope they add it to the final submission.

**Key Questions For Authors:**

1. How well does the proposed lattice representation generalize across platforms with different spatial layouts?
2. Do the predicted gene expression maps recover known spatial gene patterns or tissue structures?
3. Since the model relies on accurate coordinate alignment between H&E and ST spots, how robust is it to small coordinate misalignments?

**Limitations:**

Yes

**Strengths And Weaknesses:**

Strengths:
The submission tackles a vital problem: making spatially resolved molecular analysis cheaper and more scalable through histology. The motivation is strong, and the technical execution is sound.

The core innovation is modeling ST locations as a sparse lattice rather than a dense grid. By using a dyadic encoder-decoder with sparse local attention, the model efficiently expands its receptive field without the overhead of dense processing.

HiST also proves its worth through performance. It is significantly faster and more lightweight than STFlow while delivering better predictive accuracy. Finally, the thorough ablation study validates key design choices, such as coordinate augmentation, multiscale fusion, and the slide calibration token.

Weaknesses:
1. The obvious thing is that the novelty of the method is somewhat limited. Many components, such as hierarchical encoder–decoder design and window attention, are adapted from existing work. The main contribution appears to be combining these ideas for this application rather than introducing a fundamentally new method.

2. The authors compare HiST to ST-Net, BLEEP, UNI, and STFlow, but the selection lacks clear justification. Given the focus on sparse, irregular sampling and long-range context, stronger comparisons are needed. The paper should include benchmarks against graph-based or irregular-coordinate architectures to fully validate its claims.

3. The evaluation is dominated by (R^2) and PCC, which mainly capture spot-wise regression quality. For this application, I would encourage adding metrics that better reflect spatial and biological fidelity, such as gene-wise Spearman correlation, MAE/RMSE, agreement in spatial autocorrelation statistics (e.g., Moran’s (I)), recovery of spatially variable genes, pathway-level activity concordance, and downstream utility measures such as spatial clustering consistency, cell-type/niche localization accuracy, or neighborhood preservation.

4. Please work in typing issue for the next submission: In page 6, "The benchmark also ßincludes data across multiple ST platforms."; some missing space between citations; Equation (1) refers to Proj in the text: “and Proj denotes a learned output projection” But Proj is not shown explicitly in the equation, creating a mismatch between equation and description.

5.  Please also include these papers to discuss and position work better against them: https://doi.org/10.1038/s41592-021-01255-8,
https://doi.org/10.48550/arXiv.2507.04704, https://doi.org/10.64898/2025.12.09.693287, https://doi.org/10.1038/s41467-025-63915-z

---

> ### Author Rebuttal · Authors · 2026-03-31
>
> We sincerely thank Reviewer XWGZ for the encouraging assessment and constructive feedback. We address each point below.
>
> > **W1.** Limited novelty.
>
> Thank you for raising this concern. We acknowledge that many of HiST's individual components draw on existing ideas such as hierarchical encoder–decoder design and window attention. Our contribution lies in adapting these ideas to a problem setting where direct transfer from dense-image architectures is non-trivial: whole-slide H&E-to-ST inference over irregular, sparse measurement supports. A helpful distinction is that, in related pipelines, hierarchical backbones are typically used at the patch-level images, whereas HiST applies a sparse multiscale hierarchy at the whole-slide level over measured locations. The components are coupled through a shared sparse-support representation, which jointly enables multiscale whole-slide modeling efficiently. We believe this problem-driven integration is reflected in the substantial efficiency and accuracy gains over prior work. We will revise the paper to make this scope of contribution more explicit.
>
> > **W2 & Q1.** Baseline selection and cross-platform generalization.
>
> Thank you for raising these important points. We address both concerns together here.
>
> **Baseline selection.** Our baselines were chosen to cover main H&E-to-ST paradigms scalable to whole-slide inference: ST-Net and UNI (direct prediction), BLEEP (contrastive embedding retrieval and imputation), and STFlow (multi-step generative model). Notably, STFlow explicitly employs a kNN-graph construction with edge-conditioned message passing (analogous to GAT) over measured ST locations. Therefore, our evaluation already directly benchmarks our sparse lattice against a leading graph-based architecture. We will clarify STFlow's graph-theoretic nature and expand our literature review accordingly.
>
> **Cross-platform generalization.** Our evaluation spans multiple ST technologies with fundamentally different spatial layouts, including Visium, Xenium, and legacy ST. By design, HiST is platform-agnostic: its sparse lattice is constructed directly from each slide's observed coordinates rather than assuming a fixed platform-specific grid, making the representation naturally adaptable across these diverse layouts. Table 2 shows consistent improvements across different platforms. We will clarify the scope of our generalization claims in the revision.
>
> > **W3 & Q2.** Evaluation beyond regression metrics and recovery of spatial patterns.
>
> Thanks for this valuable suggestion. Among the metrics you recommended, we prioritized gene-wise Spearman correlation (SCC) and Moran's I correlation as they most directly assess spatial fidelity. We note that gene-wise SCC is inherently constrained across all methods: in ST data, the majority of genes are highly sparse and zero-inflated, meaning even minor prediction noise on these genes severely disrupts rank correlation and drags the score toward zero. Despite this dataset-intrinsic challenge, HiST achieves a gene-wise SCC of 0.1421 compared to STFlow's 0.1255, representing a ~13% relative improvement. HiST consistently outperforms STFlow on Moran's I correlation as well (0.497 vs. 0.476), confirming that the gains extend beyond spot-wise regression to gene-wise recovery. We further provide qualitative spatial gene maps at https://anonymous.4open.science/r/anonymous-9F20, which visually demonstrate that HiST recovers spatial gene patterns. We will explore additional downstream metrics such as spatial clustering consistency and incorporate these results in the revision.
>
> > **W4 & W5.** Issues in typography and literature coverage.
>
> Thank you for this constructive comment. We will correct the noted presentation issues, including the stray character, missing citation spaces, and the mismatch between Eq. (1) and $\mathrm{Proj}$. We also agree that the related-work discussion should better position HiST. In Sec. 2, we will expand coverage to include graph-based ST methods such as SpaGCN, multimodal/generative spatial models such as SPATIA, cross-modal integration methods such as SpatialMETA, and downstream ST analysis methods such as NicheAgent, among others, while clarifying that HiST targets efficient coordinate-guided H&E-to-ST prediction with a sparse hierarchical whole-slide backbone.
>
> > **Q3.** Robustness to coordinate misalignment.
>
> Thanks for this insightful question. Robustness to small coordinate misalignments is one of the motivations behind our coordinate augmentation strategy: during training, we inject random spatial perturbations into the lattice coordinates, forcing the model to tolerate inexact alignment. As shown in the ablation study, removing coordinate augmentation leads to a performance drop, suggesting that this strategy acts as an effective regularizer against the alignment noise present in real-world H&E and ST data. We will add a discussion of this connection in the revision.

---

> > ### Author Rebuttal · Reviewer_XWGZ · 2026-03-31
> >
> > Thank you for the detailed rebuttal and clarifications. I appreciate the authors’ effort in addressing my concerns and improving the positioning of the work.
> >
> > Regarding novelty, I understand the authors’ argument that the contribution lies in adapting hierarchical transformer components to a sparse, whole-slide lattice setting rather than introducing entirely new architectural elements. This clarification is helpful, and I agree that the problem setting makes direct reuse of dense-image designs non-trivial. However, I still view the novelty as primarily an integration and adaptation of existing ideas, rather than a fundamentally new modeling paradigm.
> >
> > On baseline selection, the clarification that STFlow incorporates graph-based message passing is useful. This partially addresses my concern about missing comparisons to irregular-coordinate or graph-based methods. That said, I still believe the paper would benefit from a more explicit and systematic comparison to a broader class of graph or coordinate-aware architectures, or at least a clearer justification of why the current set is sufficient.
> >
> > For evaluation, I appreciate the addition and discussion of gene-wise Spearman correlation and Moran’s I. These metrics strengthen the claims about spatial fidelity. However, the evaluation is still somewhat limited in capturing downstream biological utility (e.g., spatial clustering consistency, cell-type localization, or pathway-level agreement). While I understand that adding such experiments may be beyond the scope, I encourage including them in the revision if possible.
> >
> > The clarification on cross-platform generalization and robustness to coordinate misalignment is helpful and addresses my questions conceptually. The use of coordinate augmentation as a regularization mechanism is reasonable, though additional quantitative analysis would further strengthen this claim.
> >
> > Overall, the rebuttal improves clarity and partially addresses my concerns, but some limitations, particularly around novelty positioning and evaluation breadth, remain.

---

> > > ### Author Response · Authors · 2026-04-02
> > >
> > > We sincerely appreciate your thorough engagement throughout the review process. We address the remaining points below.
> > >
> > > > Downstream biological utility.
> > >
> > > Following your suggestion, we evaluated spatial clustering consistency by running Louvain clustering on predicted and ground-truth ST and measuring ARI. Across all datasets (macro-averaged), HiST achieves an ARI of 0.631 compared to STFlow's 0.601. While gene-wise SCC remains constrained due to the zero-inflated nature of ST data, the clustering results confirm that HiST captures coarse-grained spatial organization well and maintains its advantage at the level of downstream analysis. We will include this result in the revision.
> > >
> > >
> > > > Robustness to coordinate misalignment.
> > >
> > >  Removing augmentation drops R² from 0.228 to 0.190 on Brain and from 0.308 to 0.269 on Colon, confirming that the augmentation acts as an effective spatial regularizer. We agree that a systematic evaluation under varying degrees of registration offset would provide additional insight, and will add a discussion of this direction in the revision.
> > >
> > > We hope these additional results address the remaining concerns and will incorporate suggested improvements in the revised manuscript.

---

### Decision · Program_Chairs · 2026-04-30

**Decision:**

Accept (regular)

**Comment:**

The reviewers and authors had discussed the papers thoroughly and many of their concerns have been addressed by the authors. Most of the reviewers vote for acceptance.